# ALPHASAGE: STRUCTURE-AWARE ALPHA MINING VIA GFLOWNETS FOR ROBUST EXPLORATION

**Binqi Chen**♠∗  **Hongjun Ding**♡  **Ning Shen**★  **Taian Guo**♠∗

**Jinsheng Huang**♠∗  **Luchen Liu**◇  **Ming Zhang**♠†

♠School of Computer Science, National Key Laboratory for Multimedia Information Processing, PKU-Anker LLM Lab, Peking University, China  ◇Zhengren Quant, Beijing, China
♡Baruch College, City University of New York  ★Statistics, University of British Columbia
{cbq, taianguo, hjs}@stu.pku.edu.cn, hongjun.ding.baruchmfe@gmail.com,
ning.shen@stat.ubc.ca, liulc@zhengrenquant.com, mzhang_cs@pku.edu.cn

## ABSTRACT

The automated mining of predictive signals, or alphas, is a central challenge in quantitative finance. While Reinforcement Learning (RL) has emerged as a promising paradigm for generating formulaic alphas, existing frameworks are fundamentally hampered by a triad of interconnected issues. First, they suffer from reward sparsity, where meaningful feedback is only available upon the completion of a full formula, leading to inefficient and unstable exploration. Second, they rely on semantically inadequate sequential representations of mathematical expressions, failing to capture the structure that determine an alpha's behavior. Third, the standard RL objective of maximizing expected returns inherently drives policies towards a single optimal mode, directly contradicting the practical need for a diverse portfolio of non-correlated alphas. To overcome these challenges, we introduce **AlphaSAGE** (**S**tructure-**A**ware Alpha Mining via **G**enerative Flow Networks for Robust **E**xploration), a novel framework built upon three cornerstone innovations: (1) a structure-aware encoder based on Relational Graph Convolutional Network (RGCN); (2) a new framework with Generative Flow Networks (GFlowNets); and (3) a dense, multi-faceted reward structure. Empirical results demonstrate that AlphaSAGE outperforms existing baselines in mining a more diverse, novel, and highly predictive portfolio of alphas, thereby proposing a new paradigm for automated alpha mining. Our code is available at https://github.com/BerkinChen/AlphaSAGE.

## 1 INTRODUCTION

The primary objective in quantitative trading is to identify and exploit market inefficiencies, a pursuit centered on the mining of "alphas". These alphas are predictive signals, typically represented as mathematical expressions, that aim to forecast asset returns and thus serve as the cornerstone of systematic trading strategies.[1] Therefore, *alpha mining* (efficient construction of high-quality alphas) constitutes the core of quantitative research: high-quality alphas enable more accurate return forecasting, improved risk-adjusted portfolio construction, and ultimately superior excess returns.

Traditionally, alpha mining has been a manual, hypothesis-driven process. Researchers propose financial or economic hypotheses, translate them into candidate alphas, and validate their predictive power through statistical tests or backtesting. While this pipeline has led to influential discoveries such as value, momentum, and quality alphas (Kakushadze, 2016), it suffers from limited scalability and strong reliance on human intuition. With the increasing complexity of financial markets, the hypothesis-driven paradigm struggles to cope with vast, non-linear interactions in high-dimensional data, making it increasingly challenging to uncover novel and uncorrelated signals.

---

∗Internship at Zhengren Quant.
†Corresponding author.
[1]An example of alpha is shown in Figure 1.

Recent advances have motivated the shift towards automated alpha mining, where machine learning algorithms systematically search through the enormous combinatorial space of possible formulas. Early efforts often relied on Genetic Algorithm (GA) (Chen et al., 2021; Zhang et al., 2020; Cui et al., 2021), which evolves candidate formulas using mutation and crossover operators. Despite producing interpretable formulas, GA methods can be computationally inefficient and tend to converge to local optimum if mutation rate is not carefully designed. More recently, Reinforcement Learning (RL) (Yu et al., 2023; Zhu & Zhu, 2025; Zhao et al., 2024; Xu et al., 2024) has emerged as a powerful alternative, framing alpha construction as a sequential decision-making process in which an agent incrementally builds formulas. RL-based methods promise higher efficiency and scalability but also inherit several critical challenges, including reward sparsity, structural underrepresentation, and limited diversity in generated alphas.

The direct application of RL to alpha mining is generally fraught with significant obstacles that limit its efficacy. First, current methodologies often suffer from a severe "cold-start" problem, as the reward signal—typically based on an alpha's Information Coefficient (IC)—is extremely sparse (Zhao et al., 2025). Second, most existing approaches represent alpha expressions as simple sequences of tokens, often processed by sequential models such as LSTMs. Such representations fail to capture the logical and hierarchical structure inherent in formulas. Finally, the traditional RL, which is designed to maximize a singular reward function, tends to produce a relatively uniform path for alpha mining, lacking the diversity essential for constructing robust portfolios (Tang et al., 2025).

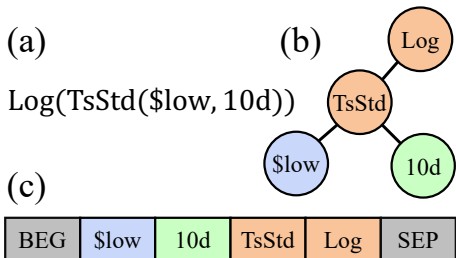

Figure 1: Different forms of alpha: (a) Formulated alpha; (b) Alpha's expression tree; (c) Reverse Polish Notation (RPN) for alpha.

To overcome these limitations, we propose **AlphaSAGE** (**S**tructure-**A**ware Alpha Mining via **G**enerative Flow Networks for Robust **E**xploration), a comprehensive framework designed to address the core problems of exploration, semantic understanding, and diversity in alpha mining. To evaluate the effectiveness of the model, we conducted extensive experiments based on real historical data from both the Chinese and U.S. stock markets. The experimental results demonstrate that the model outperforms existing models across different markets.

In summary, our contributions are as follows:

- We introduce **a structure-aware encoder** based on Relational Graph Convolutional Network (RGCN) (Schlichtkrull et al., 2018) that operates on Abstract Syntax Tree (AST) representations of alphas to capture their semantic and compositional nature.

- We propose **a generative framework** using Generative Flow Networks (GFlowNets) (Bengio et al., 2021; Malkin et al., 2022; Bengio et al., 2023) that learns to sample a diverse set of candidates, directly addressing the need for a varied alpha portfolio.

- We present **a dense, multi-faceted reward function** that combines terminal performance with intrinsic rewards for structural integrity and novelty to effectively guide the GFlowNet's exploration.

## 2    BACKGROUND AND RELATED WORK

### 2.1    ALPHAS IN QUANTITATIVE TRADING

In quantitative equity trading, an alpha is a function that maps past information about each asset, such as historical prices, volumes, or fundamental ratios, to a real valued score for every stock on a given date. At each rebalancing time $t$, the alpha values $s_{t,n}$ across stocks $n$ are typically standardized in the cross section and then transformed into a portfolio signal, for example by going long on stocks with high scores and short on stocks with low scores under a risk and leverage budget.

The next period portfolio return is

$$R_t = \sum_n w_{t,n}\, r_{t+1,n},\tag{1}$$

where $w_{t,n}$ are the portfolio weights derived from the alpha signal and $r_{t+1,n}$ are realized returns. The predictive quality of an alpha is often summarized by the information coefficient, defined as the cross sectional correlation between $s_{t,n}$ and $r_{t+1,n}$ over time. In practice, production systems maintain a library of many such alphas and combine a subset of them into a diversified trading signal, which motivates automated methods for discovering alphas with strong predictive power and low mutual correlation.

## 2.2 Alpha Mining and Combination

In quantitative finance, an *alpha* is a deterministic transformation of historical market data into a signal that aims to forecast future returns. When expressed as a symbolic program (e.g., an abstract syntax tree), an alpha remains interpretable and auditable. Alpha quality is commonly summarized by correlation-based metrics (e.g., IC) computed between alpha outputs and subsequent returns.

Early discovery pipelines were manual and hypothesis-driven. More recent automation—most notably genetic algorithm (GA) (Chen et al., 2021; Zhang et al., 2020; Cui et al., 2021) and reinforcement learning (RL) (Yu et al., 2023; Zhu & Zhu, 2025; Zhao et al., 2024; Xu et al., 2024)—expanded the search space but introduced three recurring challenges: sparse and delayed rewards, weak encoding of alpha structure, and mode collapse toward a few similar solutions. Because single alphas are typically unstable across time and markets, practitioners assemble a library of alphas and combine them into a portfolio-level signal. Simple linear combinations are prevalent in practice, yet high correlations among alphas can make coefficient estimates unreliable and reduce both robustness and interpretability.

These observations motivate frameworks that jointly optimize for predictive power and diversity. Our design follows this principle: it encourages structurally distinct alphas during generation and combines them with a transparent, adaptively weighted scheme that emphasizes low cross-alpha dependence. Additional details appear in Appendix B.1.

## 2.3 Graph Neural Networks

Graph neural networks (GNNs) (Scarselli et al., 2008; Yao et al., 2019; Schlichtkrull et al., 2018) update each node by gathering information from its neighbors and then refining the node's representation with that context. Stacking layers allows information to propagate over multiple hops, so nodes capture both local attributes and broader structural relations. When factor candidates are represented as graphs—such as trees for formulaic alphas—GNNs can encode semantic similarity and structural constraints more naturally than sequence models. This makes them attractive for learning embeddings of factors, guiding search over symbolic expressions, and measuring diversity at the representation level. Additional details appear in Appendix B.2.

## 2.4 Generative Flow Networks

Generative Flow Networks (GFlowNets) (Bengio et al., 2021; Malkin et al., 2022; Bengio et al., 2023) are generative learners that construct objects step by step and aim to sample a diverse set of high-reward solutions rather than collapsing to a single optimum. They treat generation as moving through a directed acyclic state space from an initial empty state to a terminal, valid object. By learning complementary forward and backward policies and matching "flow" through states, GFlowNets approximate a sampling distribution that is shaped by the downstream reward. Practically, this yields exploration that is both reward-aware and diversity-seeking, producing a portfolio of candidates with varied structures and competitive quality—properties that are well aligned with the needs of alpha discovery and combination. Additional details appear in Appendix B.3.

## 3 METHODOLOGY

### 3.1 FRAMEWORK OVERVIEW AND PROBLEM FORMULATION

The primary objective of automated alpha discovery is to navigate a vast, combinatorial search space $\mathcal{X}$ of potential mathematical expressions, or "alphas". Each alpha $\alpha \in \mathcal{X}$ is a function that maps historical market data for a universe of $N$ assets with $M$ features at day $d$, denoted as $X_d \in \mathbb{R}^{N \times M}$, to a vector of predictive signals $z_d = \alpha(X_d) \in \mathbb{R}^N$. The quality of these signals is evaluated against future asset returns $y_d \in \mathbb{R}^N$.

Existing RL frameworks model this as a sequential decision-making problem to construct a synergistic portfolio of alphas. In this paradigm, an agent iteratively generates new alphas to add to an evolving pool, $\mathcal{F}$. The reward for generating a new alpha, $\alpha_{\text{new}}$, is its marginal contribution to the performance of a combination model $c(\cdot)$ trained on the updated pool. The objective at each step is to find a alpha that maximizes this improvement:

$$\alpha_{\text{new}}^* = \arg \max_{\alpha \in \mathcal{X}} \mathbb{E}\left[ R(\alpha | \mathcal{F}) \right], \tag{2}$$

where the reward is defined as $R(\alpha | \mathcal{F}) = \text{IC}(c(X; \mathcal{F} \cup \{\alpha\})) - \text{IC}(c(X; \mathcal{F}))$. This formulation creates a non-stationary Markov Decision Process, as the reward for any given alpha changes whenever the pool $\mathcal{F}_t$ is updated. While this approach encourages synergy within the single, greedily constructed portfolio, it does not learn a global distribution over all high-quality alphas.

We reformulate alpha discovery as a problem of learning a **generative policy** $P_\theta(\alpha)$ that directly models the distribution of high-quality alphas over the entire space $\mathcal{X}$. The policy is trained such that the probability of sampling any alpha is proportional to a carefully designed reward function $R(\alpha)$, which reflects its intrinsic quality and novelty:

$$P_\theta(\alpha) \propto R(\alpha), \quad \forall \alpha \in \mathcal{X}, \tag{3}$$

By sampling from this learned global distribution, rather than following a single construction path, we can generate a more diverse and robust portfolio of candidate alphas.

### 3.2 ALPHA GENERATION VIA GENERATIVE FLOW NETWORKS

To address the need for a diverse portfolio, we propose a new framework with GFlowNets. A GFlowNet is a probabilistic generative model designed to learn a stochastic policy for sampling objects $\alpha$ from a space $\mathcal{X}$ with probability $P(\alpha)$ proportional to a given reward function $R(\alpha)$.

The construction of an alpha is modeled as a trajectory $\tau = (s_0 \to s_1 \to \cdots \to s_n = \alpha)$ in a state space represented as a directed acyclic graph (DAG).

- **States** ($s \in \mathcal{S}$): Partially constructed ASTs. The initial state $s_0$ is an empty tree. Terminal states are complete and valid ASTs, forming the space $\mathcal{X} \subset \mathcal{S}$.
- **Actions** ($a \in \mathcal{A}$): Adding a new token (operator or feature) to an open leaf node of a partial AST. According to the state $s$, invalid actions are masked and the next token is sampled from the masked distribution.
- **Complete Trajectories**: A full trajectory corresponds to constructing a valid expression tree. Only such trajectories are considered terminal and eligible for evaluation.

To prevent expressions from growing excessively long or from being forcefully terminated into invalid states after exceeding the maximum token length, we incorporate an **early stop** mechanism. Specifically, when the current stack already forms a valid expression, the generation process may stop with a probability:

$$p = \frac{\text{Len}(s_t)}{\text{MaxLen}}, \tag{4}$$

where $\text{Len}(s_t)$ is the number of nodes in $s_t$, and MaxLen is the maximum allowed length. This mechanism balances exploration of longer expressions with the efficiency of producing syntactically valid formulas.

A GFlowNet learns a forward policy $P_F(s_{t+1} | s_t; \theta)$ for constructing objects and a backward policy $P_B(s_t | s_{t+1}; \theta)$ for deconstruction. The training objective enforces a flow-matching condition

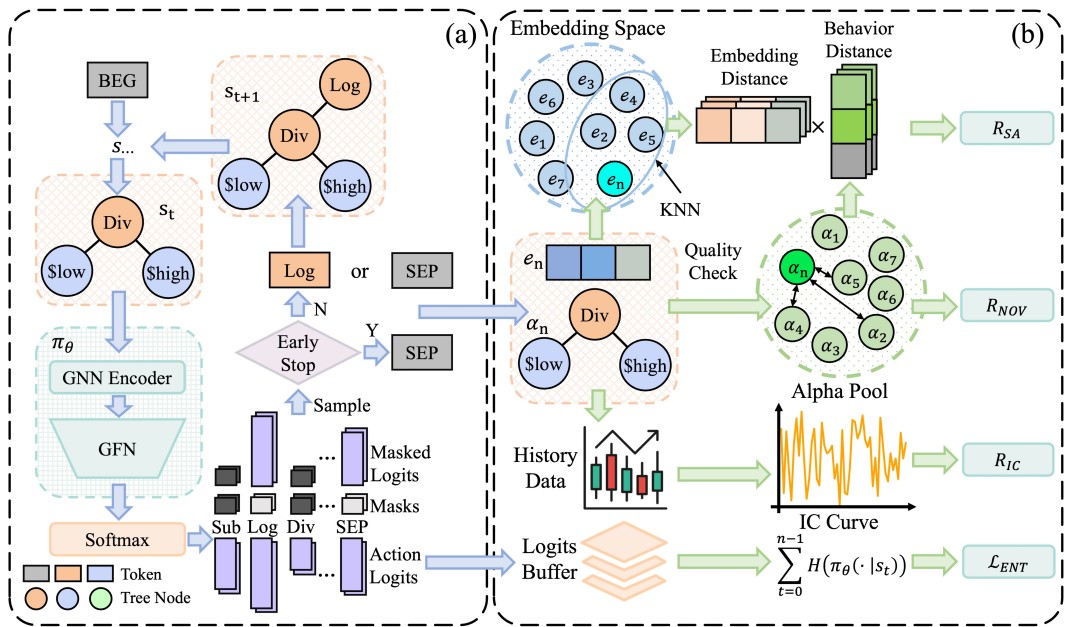

Figure 2: **An overview of AlphaSAGE. (a) AlphaGenerator.** Starting from an empty state, we iteratively construct a partial AST for the formula prefix. An RGCN encoder produces node and pooled graph embeddings, which condition the GFlowNet forward policy to sample the next to-ken from syntactically valid actions only. Rollouts terminate by emitting SEP once the minimal length is met or MaxLen is reached. The resulting formula $\alpha_n$ and its embedding $e_n$ are stored for reward evaluation and training. **(b) AlphaEvaluator.** For each sampled $\alpha_n$, we compute reward components on historical cross-sectional data: $R_{\mathrm{SA}}$ (structure alignment), $R_{\mathrm{NOV}}$ (novelty via low correlation with a reference set), and $R_{\mathrm{IC}}$ (predictive IC). We additionally use an entropy regularizer $\mathcal{L}_{\mathrm{ENT}}$ to encourage exploration. The combined reward $R(\alpha_n)$ updates the GFlowNet and encoder via the trajectory balance objective $\mathcal{L}_{\mathrm{TB}}$ augmented with $\mathcal{L}_{\mathrm{ENT}}$.

throughout the state space, ensuring that the probability of generating a complete alpha $\alpha$ matches the target distribution:

$$P(\alpha) = \sum_{\tau:s_n=\alpha} P_F(\tau) = \frac{R(\alpha)}{Z}, \tag{5}$$

where $Z = \sum_{\alpha' \in \mathcal{X}} R(\alpha')$ is a learnable parameter representing the total flow or partition function. While several loss functions exist, we use the Trajectory Balance (TB) loss, which is particularly suitable for our scenario as it focuses on full trajectories. The TB loss for a given trajectory $\tau$ is:

$$\mathcal{L}_{\mathrm{TB}}(\tau) = \left( \log Z_\theta + \sum_{t=1}^{n} \log P_F(s_t|s_{t-1};\theta) - \log R(s_n) - \sum_{t=1}^{n} \log P_B(s_{t-1}|s_t;\theta) \right)^2, \tag{6}$$

where $Z_\theta$ is a learnable scalar approximating total flow. Minimizing this loss over sampled trajectories trains the policy, resulting in a model that produces a diverse set of high-reward alphas.

### 3.3 GNN EMBEDDING AND STRUCTURE-AWARE REWARD

A fundamental limitation of existing methods is their reliance on sequential encoders (e.g., LSTMs) operating on flattened representations like Reverse Polish Notation. Such an approach fails to capture the hierarchical structure of mathematical expressions, treating logically equivalent formulas (e.g., $close + open$ and $open + close$) as different sequences. To overcome this, we first parse every formulaic alpha $\alpha$ into its corresponding AST, denoted as $\mathcal{T}_\alpha = (\mathcal{V}_\alpha, \mathcal{E}_\alpha)$, where $\mathcal{V}_\alpha$ is the set of nodes (operators and features) and $\mathcal{E}_\alpha$ is the set of edges representing the computational hierarchy. This representation is invariant to semantically inconsequential syntactic variations.

To capture the heterogeneity of relations between different types of operators and features in the $\mathcal{T}_\alpha$, we adopt RGCN as the encoder. Unlike standard GNNs that treat all edges uniformly, RGCNs explicitly model multiple relation types, which is crucial for distinguishing, for example, the edge between a temporal operator and a feature versus the edge between a temporal operator and its window length.

Each node $v \in \mathcal{V}_\alpha$ is initialized with a feature vector $h_v^{(0)}$. At layer $l$, the hidden representation of node $v$ is updated as:

$$h_v^{(l)} = \mathrm{ReLU}\left(\sum_{r\in\mathcal{R}}\sum_{u\in\mathcal{N}_r(v)}\frac{1}{c_{v,r}}W_r^{(l)}h_u^{(l-1)} + W_0^{(l)}h_v^{(l-1)}\right), \tag{7}$$

$$e_\alpha = \mathrm{MaxPooling}\left(\{h_v^{(L)}\}_{v\in\mathcal{V}_\alpha}\right), \tag{8}$$

where $\mathcal{R}$ is the set of relation types, $\mathcal{N}_r(v)$ denotes the neighbors of node $v$ connected via relation $r$, $c_{v,r}$ is a normalization constant (e.g., $|\mathcal{N}_r(v)|$), $W_r^{(l)}$ is the trainable weight matrix specific to relation $r$, and $W_0^{(l)}$ is a self-loop transformation matrix. This embedding provides a relation-aware and structure-aware representation of the alpha.

To ensure that the learned embedding is not just structurally aware but also predictive of the alpha's actual behavior, we introduce a Structure-Aware (SA) reward. The goal is to further ensure that alphas with similar structural embeddings exhibit similar behavioral patterns.

Let $Z_i \in \mathbb{R}^{D\times N}$ be the time-series vector of cross-sectionally normalized outputs for $\alpha_i$ and $Z_i(d) \in \mathbb{R}^N$ is the output at day $d$. We define a behavioral distance based on the outputs of $\alpha_i$ and $\alpha_j$:

$$d_{\mathrm{behav}}(\alpha_i, \alpha_j) = \frac{1}{D}\sum_{d=1}^{D}(Z_i(d) - Z_j(d))^2, \tag{9}$$

$$w_{ij} = \frac{\exp(-\|e_{\alpha_i} - e_{\alpha_j}\|^2)}{\sum_{k\in\mathcal{N}_K(\alpha_i)}\exp(-\|e_{\alpha_i} - e_{\alpha_k}\|^2)}, \quad j \in \mathcal{N}_K(\alpha_i), \tag{10}$$

$$R_{SA}(\alpha_i) = \exp\left(-\sum_{j\in\mathcal{N}_K(\alpha_i)}w_{ij}\cdot d_{\mathrm{behav}}(\alpha_i, \alpha_j)\right), \tag{11}$$

where $\mathcal{N}_K(\alpha_i)$ is $K$-nearest neighbors of $\alpha_i$.

## 3.4 Multi-Faceted Reward Function and Training Objective

The effectiveness of the GFlowNet is critically dependent on the design of the reward function $R(\alpha)$. To address reward sparsity and guide exploration effectively, we design a dense, multi-faceted reward function that dynamically combines several components.

The total reward for a completed $\alpha$ at training step $T$ is a weighted sum of three components:

1. **Terminal Performance Reward** ($R_{\mathrm{IC}}$): The primary measure of an alpha's predictive power, defined as its Information Coefficient:

$$R_{\mathrm{IC}}(\alpha) = \mathrm{IC}(\alpha, y) = \left|\mathbb{E}_d\left[\frac{\mathrm{Cov}(\alpha(X_d), y_d)}{\sqrt{\mathrm{Var}(\alpha(X_d))\cdot\mathrm{Var}(y_d)}}\right]\right|. \tag{12}$$

2. **Structure-Aware Reward** ($R_{SA}$): As defined in Eq. 11, this reward provides a dense signal for aligning the alpha's structural embedding with its behavior.

3. **Novelty Reward** ($R_{\mathrm{NOV}}$): To encourage the discovery of novel alphas, we introduce a novelty reward. It penalizes similarity to a dynamically updated library $\mathcal{F}_{\mathrm{known}}$ of previously discovered high-quality alphas. The definition is:

$$R_{\mathrm{NOV}}(\alpha) = 1 - \max_{\alpha'\in\mathcal{F}_{\mathrm{known}}}|\mathrm{IC}(\alpha, \alpha')|. \tag{13}$$

These reward components are combined using a time-dependent weighting scheme to balance different objectives throughout the training process. The final reward function is:

$$R(\alpha, T) = R_{\text{IC}}(\alpha) + \lambda(T)R_{\text{SA}}(\alpha) + \eta(T)R_{\text{NOV}}(\alpha), \tag{14}$$

where $\lambda(T) = (1 - \frac{T}{T_{\text{anneal}}}) \cdot \lambda_{\max}$ is a scheduling function that gradually decreases the weight of the structure-aware reward, and $\eta(T) = (1 - \frac{t}{T_{\text{anneal}}}) \cdot \eta_{\max}$ is a weight for the novelty reward.

Furthermore, to prevent premature convergence and encourage fine-grained exploration at the action level, we add a policy entropy bonus to our final training objective. The objective is to minimize the expected Trajectory Balance loss regularized by the entropy of the forward policy:

$$\mathcal{L}_{ENT} = -\mathbb{E}_{\tau \sim P_F(\tau;\theta)}\left[\sum_{t=0}^{n-1} H(\pi_\theta(\cdot|s_t))\right], \tag{15}$$

$$\mathcal{L}_{\text{final}} = \mathbb{E}_{\tau \sim P_F(\tau;\theta)}[\mathcal{L}_{\text{TB}}(\tau)] + \beta \cdot \mathcal{L}_{ENT}, \tag{16}$$

where $H(\pi_\theta(\cdot|s_t))$ is the entropy of the action selection policy at state $s_t$ and $\beta$ is a hyperparameter controlling the strength of the entropy regularization. This comprehensive objective guides AlphaSAGE to learn a generative policy that produces a diverse, novel, and highly predictive portfolio of alpha alphas.

**Compatibility of reward components.** By construction, the three reward components share the same sign for desirable behaviours. Higher predictive performance, higher structure to behaviour alignment, and higher novelty all increase the overall reward, so there is no explicit negative coupling between terms. In the alpha-mining context, $R_{\text{IC}}$ drives predictive quality, $R_{\text{SA}}$ shapes the embedding space so that structurally similar alphas behave similarly, and $R_{\text{NOV}}$ discourages redundancy with respect to the evolving factor library rather than competing with $R_{\text{IC}}$. The components are therefore designed to be compatible and to act in a complementary way during training, rather than to optimise conflicting objectives.

### 3.5 ALPHA COMBINATION

For the combination stage, we follow the approach proposed in *AlphaForge* (Shi et al., 2025a). Specifically, instead of fixing a static set of alphas, the framework performs a dynamic re-selection and linear combination of mined alphas. At each period, recently effective alphas are filtered and re-weighted through simple linear regression, yielding a time-varying "Mega-Alpha."

This design is advantageous because it adapts quickly to regime shifts while maintaining interpretability: alpha contributions remain transparent, and the portfolio avoids overfitting by discarding stale or redundant signals. Compared with complex non-linear combiners, this method offers a balance between robustness, efficiency, and explanatory clarity.

## 4 EXPERIMENTS AND RESULTS

### 4.1 EXPERIMENT SETTING

**Evaluation Metrics.** Based on prior work (Yu et al., 2023; Tang et al., 2025) and real-world trading scenarios, we employed two types of metrics for model evaluation.(1): **Correlation Metrics**, including Information Coefficient (IC), IC Information Ratio (ICIR), Rank Information Coefficient (RIC), RIC Information Ratio (RICIR); (2): **Portfolio Metrics**, including Annualized Return (AR), Maximum Drawdown (MDD), Sharpe Ratio (SR). All metrics are better when higher. Detailed definitions and backtest settings are provided in the Appendix D.1.

**Datasets.** We selected three important subsets from two major markets (Yang et al., 2020): the CSI300, CSI500 and CSI1000 in the Chinese market, and the S&P500 in the U.S. market. The split of the training set/validation set/test set for the Chinese market is defined as follows: 2010-01-01 to 2020-12-31 / 2021-01-01 to 2021-12-31 / 2022-01-01 to 2024-12-31. For the US market: 2010-01-01 to 2016-12-31 / 2017-01-01 to 2017-12-31 / 2018-01-01 to 2020-12-31.[2] Detailed hyperparameter settings are provided in Appendix D.2.

---

[2] Due to limitations in the data source, the US market data used in this study concludes on 2020-12-31.

**Baselines.** We compare AlphaSAGE with several baseline approaches: (1) Traditional machine learning methods include **MLP** (Murtagh, 1991), **LightGBM** (Ke et al., 2017), and **XGBoost** (Chen & Guestrin, 2016); (2) GA-based methods include **GP** (Chen et al., 2021); (3) RL-based methods include **AlphaGen** (Yu et al., 2023) and **AlphaQCM** (Zhu & Zhu, 2025); (4) Generative adversarial networks-based methods include **AlphaForge** (Shi et al., 2025a); (5) Large Language Model(LLM)-based methods include **AlphaAgent** (Tang et al., 2025). The details of baseline are available at Appendix D.3.

## 4.2 OVERALL PERFORMANCE

Table 1 summarizes results across CSI300/500/1000 and S&P500: AlphaSAGE ranks first on all correlation metrics, with notably higher ICIR/RICIR, and these gains translate into the best portfolio outcomes (highest annualized return, lowest drawdown, highest Sharpe).

we examined robustness with respect to random initialization. On CSI300, running multiple seeds for all methods yields consistent improvements of AlphaSAGE over strong baselines in both correlation and portfolio metrics, with the same performance ranking preserved across runs (see Appendix 12). This suggests that the empirical gains of AlphaSAGE are stable and not attributable to a particular random seed.

Figure 3 further shows that on CSI300 (2022–2024) AlphaSAGE maintains a persistent lead in cumulative returns, with smoother drawdowns, faster recoveries, and stronger rebound capture; the CSI300 index lags throughout, underscoring the value of active factor discovery and combination.

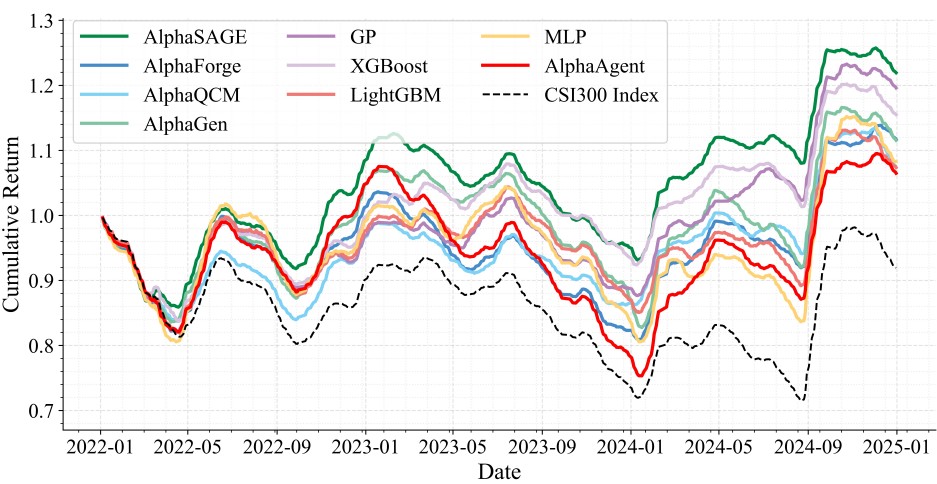

Figure 3: Cumulative return on CSI300 (2022–2024). Comparison among AlphaSAGE (ours), all baselines, and CSI300 Index benchmark.

## 4.3 ABLATION STUDY

Table 2 shows that the plain GFlowNet baseline is weakest; adding only early stopping (ES) further hurts, implying ES needs a stronger encoder. Replacing the sequence encoder with a GNN provides the largest single lift across correlation and risk metrics, underscoring the value of structure-aware representations. Adding the structure-aware reward (SA) improves ranking stability (ICIR/RICIR) and tightens drawdowns. Introducing the novelty reward (NOV) raises both signal quality and tradability by reducing redundancy among factors. Finally, the entropy regularizer (ENT) yields the best overall results—higher IC/RIC, AR, and Sharpe with controlled MDD—indicating improved exploration without brittleness and supporting the method's robustness to component choices.

## 4.4 SENSITIVITY ANALYSIS

We vary the weights of novelty reward ($R_{\text{NOV}}$) and structure-aware reward ($R_{\text{SA}}$) on CSI300 (Fig. 4). For $R_{\text{NOV}}$, correlation and portfolio metrics improve at small–moderate levels and remain on a

Table 1: Performance Comparison of Different Methods on CSI300, CSI500, CSI1000 (China) and S&P500 (U.S.). Bold and underlined numbers represent the best and second-best performance across all compared approaches, respectively.

| Dataset | Method | Correlation Metrics | | | | Portfolio Metrics | | |
|---|---|---|---|---|---|---|---|---|
| | | *IC* | *ICIR* | *RIC* | *RICIR* | *AR* | *MDD* | *SR* |
| CSI300 | MLP | 0.020 | 0.158 | 0.019 | 0.142 | 3.54% | -20.9% | 0.68 |
| | LightGBM | 0.011 | 0.124 | 0.006 | 0.064 | 2.61% | -18.5% | 0.53 |
| | XGBoost | 0.031 | 0.243 | 0.033 | 0.248 | 5.40% | -17.5% | 1.26 |
| | GP | 0.026 | 0.215 | 0.028 | 0.216 | 6.80% | -17.6% | 1.55 |
| | AlphaGen | 0.058 | 0.414 | 0.057 | 0.360 | 4.00% | -22.6% | 0.76 |
| | AlphaQCM | 0.043 | 0.262 | 0.042 | 0.246 | 1.95% | -24.8% | 0.36 |
| | AlphaForge | 0.041 | 0.259 | 0.052 | 0.306 | 3.90% | -21.9% | 0.88 |
| | AlphaAgent | 0.051 | 0.325 | 0.056 | 0.329 | 2.16% | -26.9% | 0.65 |
| | **AlphaSAGE(ours)** | **0.079** | **0.496** | **0.094** | **0.583** | **7.62%** | **-17.3%** | **1.71** |
| CSI500 | MLP | 0.017 | 0.185 | 0.020 | 0.233 | 1.56% | -24.3% | 0.27 |
| | LightGBM | 0.024 | 0.305 | 0.021 | 0.264 | 4.61% | -17.5% | 0.89 |
| | XGBoost | 0.039 | 0.365 | 0.052 | 0.528 | 5.50% | -17.1% | 1.15 |
| | GP | 0.014 | 0.238 | 0.022 | 0.233 | 3.04% | -19.4% | 0.56 |
| | AlphaGen | 0.032 | 0.270 | 0.031 | 0.230 | 1.15% | -32.4% | 0.19 |
| | AlphaQCM | 0.048 | 0.378 | 0.073 | 0.546 | 4.06% | -24.0% | 0.75 |
| | AlphaForge | 0.053 | 0.345 | 0.083 | 0.600 | 4.18% | -16.7% | 0.93 |
| | AlphaAgent | 0.053 | **0.396** | 0.065 | 0.495 | 1.82% | -22.4% | 0.36 |
| | **AlphaSAGE(ours)** | **0.054** | 0.379 | **0.084** | **0.637** | **5.53%** | **-16.0%** | **1.20** |
| CSI1000 | MLP | 0.048 | 0.384 | 0.069 | 0.621 | 3.22% | -25.7% | 0.47 |
| | LightGBM | 0.067 | 0.501 | 0.083 | 0.656 | 4.98% | -22.7% | 0.98 |
| | XGBoost | 0.062 | 0.498 | 0.086 | 0.695 | 4.72% | -23.5% | 0.91 |
| | GP | 0.058 | 0.474 | 0.079 | 0.657 | 4.32% | -24.7% | 0.67 |
| | AlphaGen | 0.071 | 0.540 | 0.092 | 0.713 | 5.27% | -24.0% | 0.92 |
| | AlphaQCM | 0.065 | 0.453 | **0.107** | 0.682 | 7.12% | -20.6% | 1.31 |
| | AlphaForge | 0.071 | 0.537 | 0.095 | **0.742** | 6.07% | -21.1% | 1.06 |
| | AlphaAgent | 0.072 | 0.579 | 0.089 | 0.712 | 5.51% | -20.5% | 1.01 |
| | **AlphaSAGE(ours)** | **0.076** | **0.582** | 0.096 | 0.736 | **7.62%** | **-20.3%** | **1.33** |
| S&P500 | MLP | 0.035 | 0.287 | 0.020 | 0.143 | 12.85% | -5.6% | 3.35 |
| | LightGBM | 0.023 | 0.196 | 0.018 | 0.165 | 11.11% | -5.1% | 4.22 |
| | XGBoost | 0.016 | 0.159 | 0.026 | 0.168 | 13.25% | -8.3% | 3.61 |
| | GP | 0.032 | 0.308 | 0.002 | 0.016 | 13.39% | -13.0% | 3.15 |
| | AlphaGen | 0.044 | 0.396 | 0.013 | 0.127 | 10.31% | -5.5% | 3.96 |
| | AlphaQCM | 0.038 | 0.262 | 0.010 | 0.071 | 13.86% | -13.0% | 3.30 |
| | AlphaForge | 0.039 | 0.422 | 0.031 | 0.324 | 17.24% | -5.0% | 6.30 |
| | AlphaAgent | 0.048 | 0.479 | 0.033 | 0.315 | 18.66% | -5.7% | 6.27 |
| | **AlphaSAGE(ours)** | **0.052** | **0.493** | **0.038** | **0.382** | **19.47%** | **-4.2%** | **6.32** |

broad plateau before tapering when novelty dominates. For $R_{\mathrm{SA}}$, improvements are largely monotonic across correlation and portfolio metrics with stable drawdowns. Overall, AlphaSAGE exhibits smooth responses without abrupt performance drops, indicating robustness to a wide range of hyperparameter choices and low sensitivity around the operating region.

## 5 CONCLUSION

We introduced AlphaSAGE, a structure-aware, diversity-seeking framework for formulaic alpha discovery and combination. The approach unifies a GNN encoder for symbolic expressions, a

Table 2: Ablation study on CSI300. The base model is GflowNets, where ES denotes Early Stopping, GNN indicates whether a GNN or LSTM is used as the encoder, SA stands for Structure-Aware Reward, NOV represents Novelty Reward, and ENT denotes Entropy Loss.

| Included Components | | | | | Correlation Metrics | | | | Portfolio Metrics | | |
|---|---|---|---|---|---|---|---|---|---|---|---|
| ES | GNN | SA | NOV | ENT | *IC* | *ICIR* | *RIC* | *RICIR* | *AR* | *MDD* | *SR* |
| ✗ | ✗ | ✗ | ✗ | ✗ | 0.048 | 0.393 | 0.057 | 0.437 | 3.63% | -22.9% | 0.72 |
| ✓ | ✗ | ✗ | ✗ | ✗ | 0.046 | 0.313 | 0.060 | 0.397 | -0.47% | -24.8% | -0.11 |
| ✓ | ✓ | ✗ | ✗ | ✗ | 0.070 | 0.495 | 0.088 | 0.554 | 5.58% | -19.4% | 1.25 |
| ✓ | ✓ | ✓ | ✗ | ✗ | 0.071 | 0.453 | 0.088 | 0.566 | 4.68% | -17.6% | 1.14 |
| ✓ | ✓ | ✓ | ✓ | ✗ | 0.075 | 0.494 | 0.092 | **0.614** | 6.77% | -17.8% | 1.53 |
| ✓ | ✓ | ✓ | ✓ | ✓ | **0.079** | **0.496** | **0.094** | 0.583 | **7.62%** | **-17.3%** | **1.71** |

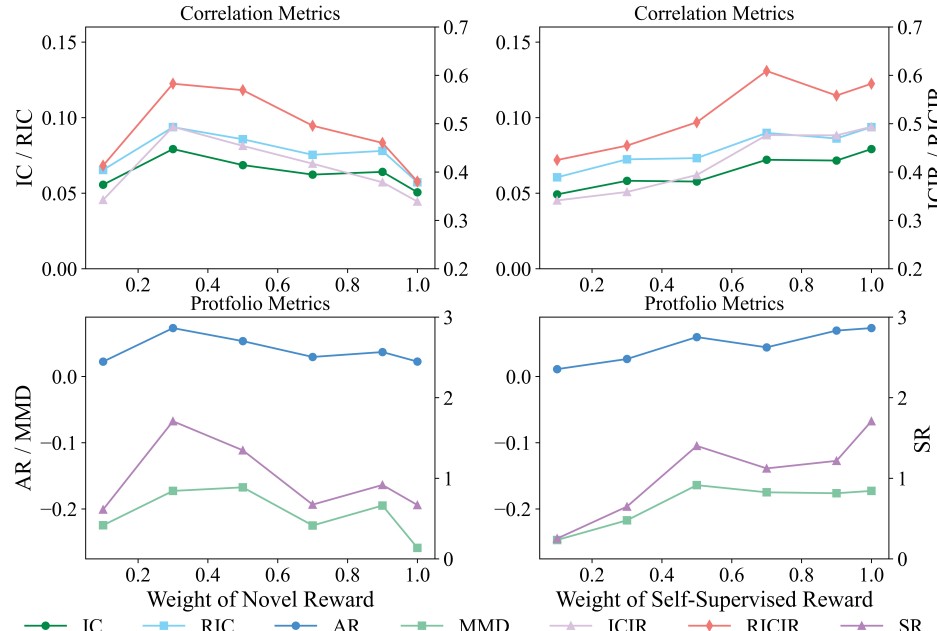

Figure 4: Sensitivity analysis of the weights for $R_{NOV}$ and $R_{SA}$ on CSI300. For the y-axis, IC, RIC, AR, and MDD refer to the axis on the left; ICIR, RICIR, and SR refer to the axis on the right.

GFlowNet generator that explores multiple high-reward modes, and a multi-signal training objective coupling predictive quality, representation–behavior alignment, novelty pressure, and entropy-based regularization. A transparent, dynamic linear combiner then translates candidate alphas into a tradable portfolio signal while maintaining interpretability.

Empirically, AlphaSAGE delivers first-rank correlation metrics across CSI300/500 and S&P500 and consistently converts these gains into superior portfolio outcomes. On CSI300 (2022–2024), its cumulative return curve maintains a persistent lead with smoother drawdowns, faster recoveries, and stronger rebound capture, underscoring robust generalization across market regimes. Ablations attribute the largest single lift to structure-aware encoding (GNN), with self-supervised alignment improving rank stability and risk control, novelty rewarding useful diversity that lifts both signal quality and tradability, and entropy regularization sharpening exploration without brittleness. Sensitivity studies show smooth responses over broad ranges of the novelty and alignment weights, indicating low tuning burden and practical robustness.

Together, these results demonstrate that coupling structure-aware representation, diversity-seeking generation, and principled multi-signal supervision yields reliable improvements in both signal quality and its conversion to realized returns, while preserving transparency in how factors are generated and combined.

## ACKNOWLEDGMENTS

This paper is partially supported by grants from the National Key Research and Development Program of China with Grant No. 2023YFC3341203 and the National Natural Science Foundation of China (NSFC Grant Number 62276002).

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

# A    THE USE OF LARGE LANGUAGE MODELS (LLMS)

We declare that the use of large language models (LLMs) during the drafting of this manuscript was confined to language-related assistance, such as sentence refinement and grammatical corrections. All substantive content was independently authored by the authors and underwent rigorous review and verification following any modifications based on LLM assistance. This research did not involve any other processes reliant upon large language models.

# B    SUPPLEMENTARY BACKGROUND ON RELATED WORK

## B.1    ALPHA MINING AND COMBINATION

**Search paradigms.**    Early work emphasized manual, hypothesis-driven construction of formulaic alphas; automated search later expanded with genetic algorithms (mutation/crossover over expression trees) (Chen et al., 2021; Zhang et al., 2020; Cui et al., 2021) and reinforcement learning (sequential decision-making over token spaces) (Yu et al., 2023; Zhu & Zhu, 2025; Zhao et al., 2024; Xu et al., 2024). In addition, there are also approaches based on Large Language Models (LLMs) (Cao et al., 2025; Shi et al., 2025b; Tang et al., 2025; Li et al., 2024; Chen, 2023; Ren et al., 2025) that generate alphas using LLMs or refine existing alphas .

**Combination and multicollinearity.**    Given a library $\{z_{i,t}^{(k)}\}_{k=1}^{K}$, linear combination remains prevalent for transparency:

$$s_{i,t} = \sum_{k=1}^{K} w_k\, z_{i,t}^{(k)}, \qquad \mathbf{w} \in \mathbb{R}^K. \tag{17}$$

However, high cross-alpha correlation inflates estimator variance. Regularization and constraints mitigate this:

$$\min_{\mathbf{w}} \sum_t \left\| \mathbf{y}_{t+\Delta} - Z_t \mathbf{w} \right\|_2^2 + \lambda_2 \|\mathbf{w}\|_2^2 + \lambda_1 \|\mathbf{w}\|_1 \quad \text{s.t. } \mathbf{1}^\top \mathbf{w} = 1, \ \ \|\mathbf{w}\|_0 \le s, \tag{18}$$

where $Z_t = [z_{\cdot,t}^{(1)}, \ldots, z_{\cdot,t}^{(K)}]$ stacks alpha columns, and optional constraints control turnover or exposure. Diagnostics such as condition number or VIF help monitor collinearity. Beyond static weights, practice often uses rolling or regime-conditioned reweighting.

## B.2    GRAPH NEURAL NETWORKS (GNNS)

**Message passing view.**    A broad class of GNNs can be written as

$$\mathbf{m}_{u\to v}^{(\ell)} = \phi_{\mathrm{msg}}^{(\ell)}\big(\mathbf{h}_u^{(\ell)}, \mathbf{h}_v^{(\ell)}, \mathbf{e}_{uv}\big), \qquad \mathbf{a}_v^{(\ell)} = \square_{u\in\mathcal{N}(v)}\mathbf{m}_{u\to v}^{(\ell)}, \qquad \mathbf{h}_v^{(\ell+1)} = \phi_{\mathrm{upd}}^{(\ell)}\big(\mathbf{h}_v^{(\ell)}, \mathbf{a}_v^{(\ell)}\big), \tag{19}$$

where $\square$ is a permutation-invariant aggregator (sum/mean/max or attention). Classical instances include GCN (Yao et al., 2019), GraphSAGE (Hamilton et al., 2017), GAT (Veličković et al., 2017), GIN (Xu et al.), the MPNN family (Gilmer et al., 2017), and relational/heterogeneous variants (R-GCN) (Schlichtkrull et al., 2018).

**Spectral perspective (GCN).**    Let $\hat{A} = A + I$ and $\hat{D} = \mathrm{diag}(\sum_j \hat{A}_{ij})$. The layerwise propagation is

$$H^{(\ell+1)} = \sigma\big(\hat{D}^{-1/2}\hat{A}\hat{D}^{-1/2}H^{(\ell)}W^{(\ell)}\big), \tag{20}$$

interpretable as a low-pass filter on the graph. Repeated smoothing risks *over-smoothing*, where node embeddings become indistinguishable; residual connections, normalization, and careful depth mitigate this (Bronstein et al., 2021).

**Attention and heterogeneity.**    GAT computes attention weights $\alpha_{uv}$ over neighbors:

$$\alpha_{uv} = \frac{\exp\big(\mathrm{LeakyReLU}\big(\mathbf{a}^\top[W\mathbf{h}_u \,\|\, W\mathbf{h}_v]\big)\big)}{\sum_{w\in\mathcal{N}(v)} \exp\big(\mathrm{LeakyReLU}\big(\mathbf{a}^\top[W\mathbf{h}_w \,\|\, W\mathbf{h}_v]\big)\big)}, \qquad \mathbf{h}_v' = \sigma\Big(\sum_{u\in\mathcal{N}(v)} \alpha_{uv}\, W\mathbf{h}_u\Big). \tag{21}$$

R-GCN introduces relation-specific parameters:

$$\mathbf{h}_v^{(\ell+1)} \;=\; \sigma\Big( \sum_{r \in \mathcal{R}} \sum_{u \in \mathcal{N}_r(v)} \frac{1}{c_{v,r}} W_r^{(\ell)} \mathbf{h}_u^{(\ell)} \;+\; W_0^{(\ell)} \mathbf{h}_v^{(\ell)} \Big). \tag{22}$$

**Expressivity and readout.** GIN links message passing to Weisfeiler–Lehman tests and uses a sum-aggregation MLP to approach maximal discriminative power in the 1-WL regime (Xu et al.). Graph-level outputs use readouts

$$\mathbf{h}_G \;=\; \mathrm{READOUT}\big( \{\mathbf{h}_v^{(L)}\}_{v \in G} \big), \quad \text{e.g., sum/mean/max.} \tag{23}$$

Positional or structural encodings (e.g., Laplacian eigenvectors, distance encodings) can further enhance global awareness (Li et al., 2020). Practical training relies on sampling and partitioning for scale (Chiang et al., 2019; Zheng et al., 2020), with OGB benchmarks standardizing evaluation (Hu et al., 2020).

**Over-smoothing and over-squashing.** Deep stacks may over-smooth; curvature-inspired rewiring, residuals, and normalization layers are common responses. Over-squashing—the compression of exponentially many distant signals into fixed-size messages—can be alleviated by attention/edge weighting, graph rewiring, and subgraph-based encoders (Bronstein et al., 2021).

### B.3 GENERATIVE FLOW NETWORKS (GFLOWNETS)

**Objective: sampling proportional to reward.** Given a set of terminal objects $\mathcal{X}$ and a non-negative reward $R : \mathcal{X} \to \mathbb{R}_{\geq 0}$, GFlowNets seek a policy that samples $x \in \mathcal{X}$ with

$$P_\theta(x) \;\propto\; R(x), \qquad P_\theta(x) \;=\; \sum_{\tau \in \mathcal{T}(x)} P_\theta(\tau), \tag{24}$$

where $\tau = (s_0 \to \cdots \to x)$ is a trajectory in a DAG of states and $\mathcal{T}(x)$ is the set of trajectories ending at $x$ (Bengio et al., 2021; 2023).

**Detailed-balance (DB) and trajectory-balance (TB).** Let $F_\theta(s) > 0$ denote a learnable *flow* through state $s$. DB enforces local conservation:

$$F_\theta(s)\, P_\theta(s' \,|\, s) \;=\; F_\theta(s')\, P_\theta(s \,|\, s') \quad \text{for edges } s \leftrightarrow s'. \tag{25}$$

TB provides a path-wise condition linking forward/backward policies and a scalar $Z_\theta$ (partition function):

$$\mathcal{L}_{\mathrm{TB}} \;=\; \mathbb{E}_\tau \Big[ \big( \log P_\theta(\tau) + \log Z_\theta - \log R(x) \big)^2 \Big], \tag{26}$$

encouraging $P_\theta(x) \propto R(x)$ when minimized (Malkin et al., 2022). Subtrajectory balance (SubTB) generalizes TB to partial paths for credit assignment (Madan et al., 2023).

**Forward/backward policies and partition function.** A typical parameterization factors $P_\theta(\tau) = \prod_{t=0}^{T-1} P_\theta(s_{t+1} \,|\, s_t)$, $P_\theta(s_t \,|\, s_{t+1})$ learned for DB/SubTB, and treats $Z_\theta$ as a learnable scalar (or function) estimating $\sum_x R(x)$. Estimation stability can be improved via baselines, variance reduction, and regularization.

**Mode coverage vs. RL/EBM/MCMC.** Unlike standard RL objectives that often favor a single high-return mode under sparse rewards, GFlowNets learn a distribution covering *multiple* modes. Compared to energy-based models (EBMs) and MCMC, GFlowNets amortize sampling via learned policies, reducing the need for long chains while retaining a reward-shaped target (LeCun et al., 2006; Salimans et al., 2015; Bengio et al., 2023). Empirical applications span molecular design, program synthesis, and discrete structure generation (Zhang et al., 2023; Jain et al., 2022), with ongoing work on offline training, replay buffers, and credit assignment (Lahlou et al., 2023).

## C  IMPLEMENTATION DETAILS

### C.1  PSEUDO CODE

The pseudo code of AlphaSAGE (core of mining framework) is shown in Algorithm 1. And the code is available at https://github.com/BerkinChen/AlphaSAGE.

---

**Algorithm 1:** AlphaSAGE

---

**Input:** Stock features $X$, stock trend labels $y$, action set $\mathcal{A}$
**Output:** Final alpha pool $\mathcal{F}$

1  **Initialize:** GFN parameters $\theta$; probability buffer $\mathcal{B}_{\text{prob}} \leftarrow \varnothing$; embedding buffer $\mathcal{B}_{\text{emb}} \leftarrow \varnothing$; alpha pool $\mathcal{F} \leftarrow \varnothing$;
2  **for** *step* $t = 1, 2, \ldots, T_{\max}$ **do**
3      Parse current state $s_t \rightarrow \texttt{ast}_t$;
4      Compute state embedding $e_t \leftarrow f_{\text{GNN}}(\texttt{ast}_t)$;           // Eq. 8
5      Output action distribution $\pi_\theta(\cdot \mid \texttt{ast}_t)$; append to $\mathcal{B}_{\text{prob}}$;
6      **if** *rand()* $\geq p_{es}(s_t)$ **then**
7          $a_t \leftarrow \texttt{SEP}$;
8      **else**
9          Sample or select $a_t \in \mathcal{A}$ from $\pi_\theta(\cdot \mid \texttt{ast}_t)$;
10     **if** $a_t = SEP$ **then**
11         Build expression $\alpha \leftarrow \texttt{BuildExpr}(s_t)$;
12         Compute alpha $z \leftarrow \texttt{ComputeAlpha}(\alpha, X)$;
13         Compute correlation reward $R_{\text{IC}} \leftarrow \texttt{IC}(z, y)$;       // Eq. 12
14         Compute novelty to pool members $R_{\text{NOV}} \leftarrow \texttt{Novelty}(z, \mathcal{F})$;     // Eq. 13
15         Run KNN on terminal embedding $e_t$ against pool embeddings: $\mathcal{N} \leftarrow \texttt{KNN}(e_t, \texttt{Emb}(\mathcal{F}), k)$;
16         Build distance-weight matrix $W \leftarrow \texttt{Dist2Weight}(\mathcal{N})$ and performance similarity $\texttt{sim} \leftarrow \texttt{PerfSim}(s, \mathcal{N}, W)$;
17         SA reward $R_{\text{SA}} \leftarrow \texttt{SAReward}(\texttt{sim})$;       // Eq. 11
18         Total reward $R \leftarrow \texttt{Combine}(R_{\text{IC}}, R_{\text{NOV}}, R_{\text{SA}}, t)$;       // Eq. 14
19         **if** *PassThreshold*$(R)$ **then**
20             $\mathcal{F} \leftarrow \mathcal{F} \cup \{\alpha\}$;
21             Append $e_t$ to $\mathcal{B}_{\text{emb}}$;
22         Trajectory Balance loss $\mathcal{L}_{\text{TB}} \leftarrow \texttt{TrajectoryBalance}(\mathcal{B}_{\text{prob}}, R)$;    // Eq. 6
23         Entropy regularizer $\mathcal{L}_{\text{ent}} \leftarrow \texttt{EntropyReg}(\mathcal{B}_{\text{prob}})$; clear $\mathcal{B}_{\text{prob}}$;     // Eq. 15
24         Total loss $\mathcal{L}_{final} \leftarrow \mathcal{L}_{\text{TB}} + \lambda_{\text{ent}} \mathcal{L}_{\text{ent}}$;       // Eq. 16
25         Update $\theta \leftarrow \theta - \eta \nabla_\theta \mathcal{L}_{final}$;
26         Reset $S_{t+1} \leftarrow \texttt{InitState}()$; clear $\mathcal{B}_{\text{prob}}$;
27     **else**
28         State transition $S_{t+1} \leftarrow \texttt{Transition}(S_t, a_t)$;
29 **return** $\mathcal{F}$;

---

### C.2  RELATION TYPE OF RGCN

To denote combinations between different operators and features, we have defined the following relationships: ① Unary operator with operand; ② Commutative operator with operands; ③ Non-commutative operator with left operand; ④ Non-commutative operator with right operand; ⑤ Rolling operator with feature operand; ⑥ Rolling operator with time operand.

### C.3  FEATURES AND OPERATORS

All operators and features available during alpha mining are listed in the Table 3.

Table 3: Raw features and operators. **F**: base market features; **U/B**: unary/binary operators; **CS**: cross-sectional operation (within-day across assets); **TS**: time-series operation (rolling window). The lookback length $d$ denotes the past $d$ trading days, and the $\epsilon$ is used only for numerical stability.

| Name | Type | Description |
|------|------|-------------|
| Open | F | Opening price |
| Close | F | Closing price |
| High | F | Daily highest price |
| Low | F | Daily lowest price |
| Vwap | F | Daily average price, weighted by the volume of trades at each price |
| Volume | F | Trading volume (number of shares) |
| Abs | U | Absolute value of the input |
| Slog1p | U | Signed log transform: sign(input) times log of (1 plus the absolute value) |
| Inv | U | Reciprocal of the input; add $\epsilon$ to avoid division by zero |
| Sign | U | Sign of the input, returning -1, 0, or 1 |
| Log | U | Natural logarithm of the input; add $\epsilon$ for numerical stability |
| Rank | U-CS | Cross-sectional rank normalization within a day, mapped to the range [0, 1] |
| Add | B | Element-wise addition of two inputs |
| Sub | B | Element-wise subtraction: first minus second |
| Mul | B | Element-wise multiplication |
| Div | B | Element-wise division; add a small constant to the denominator for stability |
| Pow | B | Element-wise power: raise the first input to the power of the second |
| Greater | B | Element-wise comparison: 1 if first input is greater than second, else 0 |
| Less | B | Element-wise comparison: 1 if first input is less than second, else 0 |
| Ref | U-TS | Lag operator: the value from d days ago |
| TsMean | U-TS | Rolling mean over the past d days |
| TsSum | U-TS | Rolling sum over the past d days |
| TsStd | U-TS | Rolling standard deviation over the past d days |
| TsIr | U-TS | Rolling information ratio over the past d days |
| TsMinMaxDiff | U-TS | Rolling range over the past d days (rolling max minus rolling min) |
| TsMaxDiff | U-TS | Current value minus the rolling max over the past d days |
| TsMinDiff | U-TS | Current value minus the rolling min over the past d days |
| TsVar | U-TS | Rolling variance over the past d days |
| TsSkew | U-TS | Rolling skewness over the past d days |
| TsKurt | U-TS | Rolling kurtosis over the past d days |
| TsMax | U-TS | Rolling maximum over the past d days |
| TsMin | U-TS | Rolling minimum over the past d days |
| TsMed | U-TS | Rolling median over the past d days |
| TsMad | U-TS | Rolling median absolute deviation over the past d days |
| TsRank | U-TS | Rolling rank of the current value within the past d days, mapped to [0, 1] |
| TsDelta | U-TS | Change over d days: current value minus the value d days ago |
| TsDiv | U-TS | Ratio over d days: current value divided by the value d days ago |
| TsPctChange | U-TS | Percentage change over the past d days |
| TsWMA | U-TS | Linearly decaying weighted moving average over the past d days |
| TsEMA | U-TS | Exponential moving average with a decay over the past d days |
| TsCov | B-TS | Rolling covariance between two inputs over the past d days |
| TsCorr | B-TS | Rolling Pearson correlation between two inputs over the past d days |

# D    EXPERIMENT DETAILS

## D.1    METRIC DETAILS

For all evaluation metrics, we provide definitions and brief interpretations. Let

$$\rho_d = \frac{\mathrm{Cov}(\alpha(X_d),\, y_d)}{\sqrt{\mathrm{Var}(\alpha(X_d))\,\mathrm{Var}(y_d)}} \quad \text{(cross-sectional correlation on day } d\text{)},\tag{27}$$

and let $R_d$ denote the portfolio return constructed from $\alpha$ on day $d$, $K$ the number of periods per year (e.g., $K{=}252$ for daily), $r_{f,d}$ the risk-free rate, and

$$W_t = \sum_{u \leq t}\bigl(1 + R_u\bigr) \quad \text{(cumulative wealth).}\tag{28}$$

- **Information Coefficient (IC)**: See Eq. 12. *Interpretation.* Cross-sectional predictive power of the factor—how well $\alpha(X_d)$ aligns with next-period outcomes $y_d$. Using the absolute value isolates *magnitude* rather than sign (long/short direction can be flipped). Higher IC indicates more informative date-wise rankings and is a prerequisite for constructing profitable long–short portfolios.

- **Information Ratio of IC (ICIR)**:

$$\mathrm{ICIR} = \frac{\mathbb{E}_d[\rho_d]}{\sqrt{\mathrm{Var}_d(\rho_d)}}.\tag{29}$$

  *Interpretation.* Time-series consistency of cross-sectional predictability: mean IC relative to its volatility. Under weak dependence, ICIR approximates a signal-to-noise measure (akin to a $t$-statistic for $\mathbb{E}[\rho_d]$), favoring factors that work *consistently* rather than sporadically.

- **Rank Information Coefficient (RankIC)**:

$$\mathrm{RankIC} = \bigl|\,\mathbb{E}_d\bigl[\rho_d^{\mathrm{rank}}\bigr]\,\bigr|, \qquad \rho_d^{\mathrm{rank}} = \frac{\mathrm{Cov}(\mathrm{rank}(\alpha(X_d)),\, \mathrm{rank}(y_d))}{\sqrt{\mathrm{Var}(\mathrm{rank}(\alpha(X_d)))\,\mathrm{Var}(\mathrm{rank}(y_d))}}.\tag{30}$$

  *Interpretation.* Spearman-style counterpart to IC that evaluates whether higher-ranked signals correspond to higher-ranked outcomes. RankIC is robust to outliers and monotone transforms of $\alpha$, aligning with rank-based portfolio constructions.

- **Information Ratio of RankIC (RankICIR)**:

$$\mathrm{RankICIR} = \frac{\mathbb{E}_d\bigl[\rho_d^{\mathrm{rank}}\bigr]}{\sqrt{\mathrm{Var}_d\bigl(\rho_d^{\mathrm{rank}}\bigr)}}.\tag{31}$$

  *Interpretation.* Time-series stability of rank-based predictive power, prioritizing factors whose cross-sectional ordering remains reliable across time.

- **Annualized Return (AR)**:

$$\mathrm{AR} = K \cdot \mathbb{E}_d[R_d].\tag{32}$$

  *Interpretation.* Economic value produced by the portfolio rule induced by $\alpha$. When compounding is material, geometric annualization via $W_t$ is preferred.

- **Maximum Drawdown (MDD)**:

$$\mathrm{MDD} = -\max_t\left(1 - \frac{W_t}{\max_{u \leq t} W_u}\right).\tag{33}$$

  *Interpretation.* Worst peak-to-trough loss of the wealth process; a trajectory- and tail-risk metric not captured by variance alone. It is critical for leverage, risk limits, and investor experience.

- **Sharpe Ratio (SR)** (annualized, excess over risk-free):

$$\mathrm{SR} = \frac{\sqrt{K}\,\mathbb{E}_d[R_d - r_{f,d}]}{\sqrt{\mathrm{Var}_d(R_d - r_{f,d})}}.\tag{34}$$

  *Interpretation.* Risk-adjusted return per unit of volatility for the $\alpha$-induced portfolio, enabling fair comparison across methods, universes, and rebalancing frequencies.

**Reporting conventions.** (i) We report IC/RankIC in absolute value (cf. Eq. 12, Eq. 30) because factor signs are arbitrary up to inversion. (ii) For SR, we use excess returns $R_d - r_{f,d}$; To ensure fair comparison while simplifying the process, we set $r_{f,d}=0$ for comparability and state this choice explicitly. (iii) Due to differing rules between the CSI300/500 indices and the S&P500, when backtesting on the CSI300/500, we purchase the top 20% of stocks each trading day and sell them after 20 days (long positions only); For the S&P500, we purchase the top 10% of stocks each trading day and sell them after 20 days, while simultaneously selling the bottom 10% of stocks and repurchasing them after 20 days (long-short combination).

## D.2 HYPERPARAMETER SETTING

The hyperparameter settings of AlphaSAGE are listed in Table 4.

Table 4: The hyperparameter settings of AlphaSAGE.

| Name | Description | Value |
|---|---|---|
| Max Length | The maximum number of tokens in the $s_t$ | 20 |
| Hidden Dim | The dimension of hidden state | 128 |
| Encoder Layer | The number of layers in RGCN encoder | 2 |
| Entropy Coef | The weight of $\mathcal{L}_{ENT}$ | 0.01 |
| Learning Rate | The learning rate to optimize $\theta$ | 0.0001 |
| SA Weight | The weight of $R_{SA}$ | 1.0 |
| NOV Weight | The weight of $R_{NOV}$ | 0.3 |
| Pool Capacity | The maximum number of alphas in alpha pool | 50 |
| Episodes | The number of trajectory sampling instances | 10000/20000[3] |

## D.3 BASELINE DETAILS

We selected seven methods as the baseline:

- **MLP** (Murtagh, 1991): A feedforward neural network that maps tabular features to return targets, capturing nonlinear interactions. It is a strong generic baseline but can overfit without careful regularization and offers limited interpretability.

- **LightGBM** (Ke et al., 2017): A gradient-boosted decision tree learner with histogram-based splits and leaf-wise growth, well suited to large, sparse, or heterogeneous financial features. It trains fast and handles missing values natively, though leaf-wise growth can overfit small samples without constraints.

- **XGBoost** (Chen & Guestrin, 2016): Boosted trees optimized with second-order information, shrinkage, column subsampling, and explicit regularization. Reliable on tabular alpha features, but the ensemble remains hard to interpret structurally and can be sensitive to label leakage or distribution shift.

- **GP** (Chen et al., 2021): Genetic programming performs symbolic regression by evolving expression trees via mutation and crossover, yielding human-readable formulas. It explores large search spaces but is prone to bloat.

- **AlphaGen** (Yu et al., 2023): Proposes mining *synergistic* sets of formulaic alphas by directly optimizing the downstream combination model's performance. Uses reinforcement learning to explore the expression search space, assigning the improvement in portfolio/combiner performance as the RL return so the generator preferentially discovers alphas that work well together.

- **AlphaQCM** (Zhu & Zhu, 2025): Frames synergistic alpha discovery as a non-stationary, reward-sparse MDP and adopts a *distributional* RL approach. Learns both a Q-function and quantiles, then applies a quantiled conditional moment method to obtain an unbiased variance estimate; the learned value and variance jointly guide exploration under non-stationarity, improving search efficiency on large universes.

- **AlphaForge** (Shi et al., 2025a): Introduces a two-stage framework that couples a genera­tive–predictive neural module for factor proposal (encouraging broad, diverse exploration) with a dynamic combination stage. The combiner selects by recent performance and *adapts weights over time*, addressing inconsistency and rigidity of fixed-weight ensembles and yielding stronger portfolio results in empirical tests.

- **AlphaAgent** (Tang et al., 2025): A multi-agent factor discovery framework is proposed. First, the Ideal Agent generates alpha ideals based on market observations and financial knowledge derived from inputs. Subsequently, the Factor Agent generates computable alpha expressions. Finally, the Eval Agent assesses the alphas, feeding the feedback back into the Ideal Agent for refinement.

For AlphaGen[4], AlphaQCM[5], AlphaForge[6] and AlphaAgent[7], we follow the default setting in their open-source projects. And for other baselines, we follow the setting in https://github.com/DulyHao/AlphaForge.

# E ADDITIONAL RESULTS

## E.1 BACKTESTING RESULTS

Figure 5 shows that on **CSI500** (2022–2024) AlphaSAGE delivers the strongest end-period wealth and sustains a clear lead for most of the horizon. It experiences *smoother drawdowns* around mid-2023, *recovers earlier* from late-2024 stress, and *retains* more of the subsequent rally; all baselines trail, while the CSI500 index lags markedly throughout.

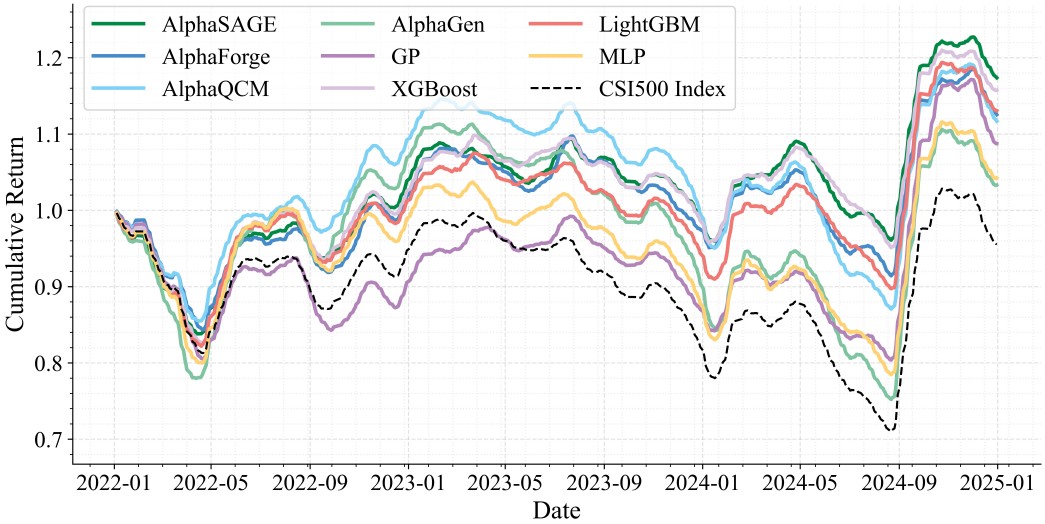

Figure 5: Cumulative return on CSI500 (2022–2024). Comparison among AlphaSAGE (ours), all baselines, and CSI500 Index benchmark.

Figure 6 shows that on **S&P500** (2018–2021) AlphaSAGE tracks near the top during calm phases, then *recovers faster* and *compounds higher* after the 2020 drawdown, finishing with the best cu­mulative return. Several baselines (e.g., AlphaForge) are competitive early but fail to match the late-period acceleration; the market index remains below AlphaSAGE by the end.

---

[4] https://github.com/RL-MLDM/alphagen
[5] https://github.com/ZhuZhouFan/AlphaQCM
[6] https://github.com/DulyHao/AlphaForge
[7] https://github.com/RndmVariableQ/AlphaAgent

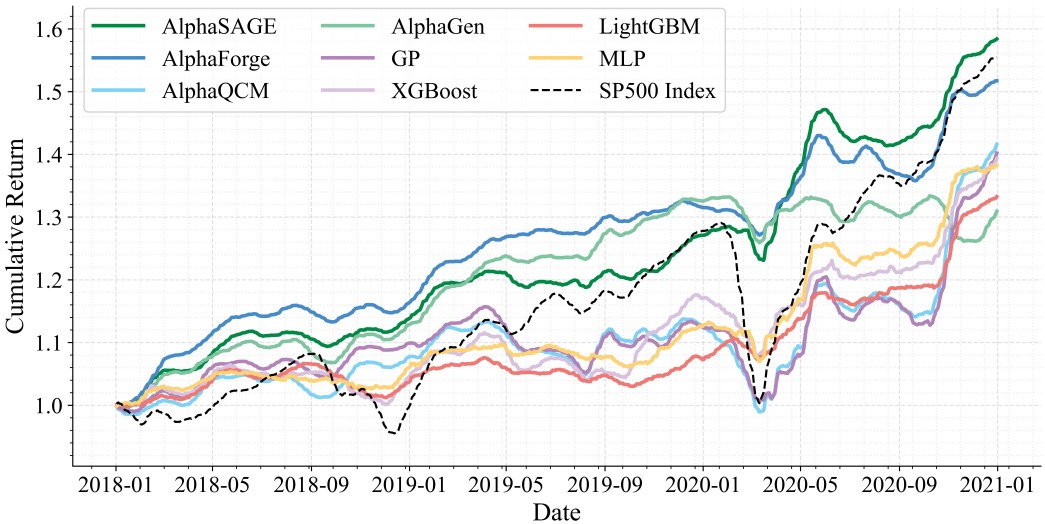

Figure 6: Cumulative return on S&P500 (2018–2020). Comparison among AlphaSAGE (ours), all baselines, and S&P500 Index benchmark.

## E.2  PARAMETER ANALYSIS

This appendix reports five practical knobs: *training steps*, *candidate-pool size*, *Max Length*, *Hidden Dim* and *Encoder Layer*. We track correlation metrics (IC/RIC, ICIR/RICIR) and portfolio metrics (AR, MDD, SR).

**Training steps.**  As shown in Fig. 7, **GFlowNets converge faster and train more efficiently than PPO** in the alpha-mining setting. IC/RIC and ICIR/RICIR rise sharply in early iterations and reach a high, stable plateau with lower variance; PPO improves more slowly and exhibits larger oscillations throughout.

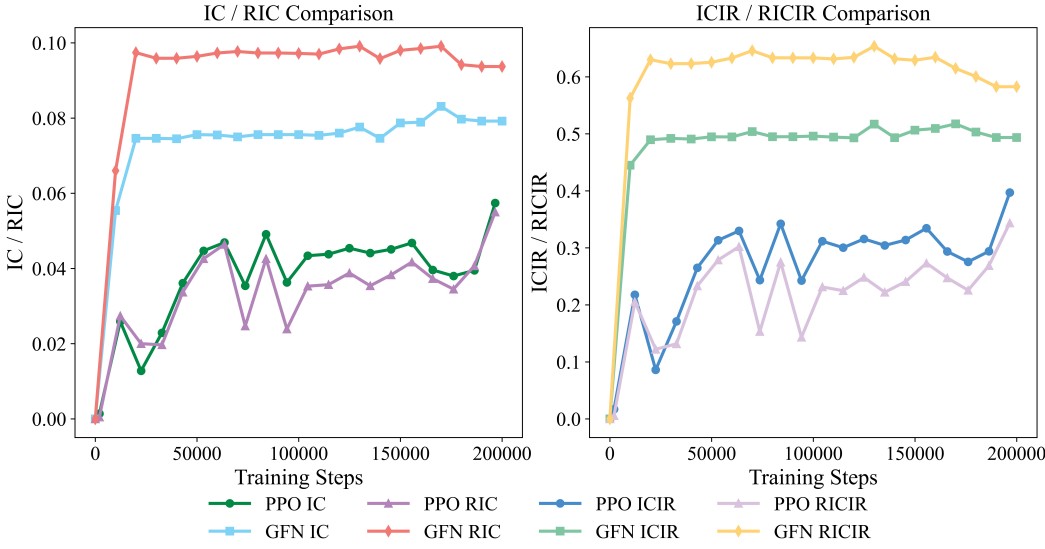

Figure 7: **Learning dynamics vs. training steps.** GFlowNets (GFN) achieve higher plateaus earlier and with less volatility than PPO across IC/RIC and ICIR/RICIR.

**Candidate-pool size.** Figure 8 shows that **increasing the factor pool yields rapid gains followed by saturation**. All correlation metrics (IC/RIC, ICIR/RICIR) and portfolio metrics (AR, SR) improve markedly when moving from very small to moderate pool sizes, then flatten into broad plateaus; MDD improves monotonically with no instability.

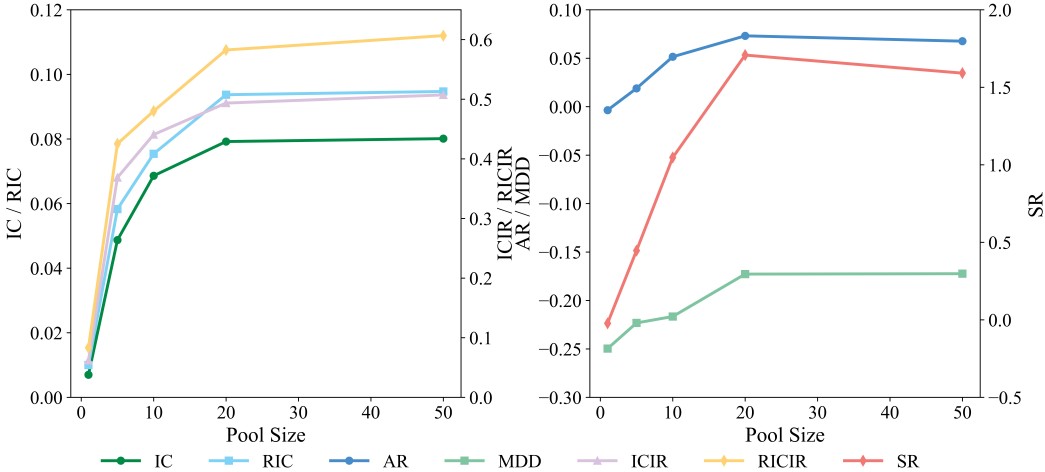

Figure 8: **Effect of candidate-pool size.** Metrics increase quickly at small–moderate pool sizes and then stabilize, indicating diminishing returns beyond a modest pool.

**MaxLen (maximum AST length).** We further study the effect of `MaxLen` in Eq. 4, which controls the maximum size of generated abstract syntax trees and thus the early-stop behaviour of the sampler. Table 5 reports mean AST depth and performance on CSI300 for different values of `MaxLen`. We observe a clear trade-off: **very small MaxLen restricts expressiveness and yields weak IC and portfolio metrics, performance improves steadily up to MaxLen = 20, and larger values hurt performance despite deeper trees**. This pattern suggests that overly long expressions introduce noise and make optimization harder, while a moderate cap (around 20 in our setting) balances expressiveness and stability.

Table 5: Sensitivity to MaxLen on CSI300. MaxLen controls the maximum AST length (and thus early-stop threshold). Mean AST depth is measured at convergence.

| MaxLen | Mean depth | IC | ICIR | RIC | RICIR | AR | MDD | SR |
|--------|-----------|-------|-------|-------|-------|--------|--------|------|
| 5 | 2.02 | 0.031 | 0.209 | 0.041 | 0.269 | -0.66% | -27.3% | 0.14 |
| 10 | 3.36 | 0.037 | 0.253 | 0.040 | 0.264 | -1.95% | -27.6% | 0.43 |
| 15 | 3.68 | 0.059 | 0.391 | 0.072 | 0.493 | 1.31% | -23.6% | 0.31 |
| 20 | 4.06 | 0.079 | 0.496 | 0.094 | 0.583 | 7.62% | -17.3% | 1.71 |
| 30 | 5.18 | 0.043 | 0.286 | 0.056 | 0.369 | -1.65% | -25.6% | 0.41 |
| 40 | 5.66 | 0.031 | 0.245 | 0.035 | 0.270 | -0.92% | -28.9% | 0.21 |
| 50 | 5.78 | 0.042 | 0.288 | 0.048 | 0.322 | 1.22% | -25.4% | 0.27 |

**Encoder Layer.** We next vary the number of RGCN layers in the structure-aware encoder on CSI300. As summarized in Table 6, **performance is stable across depths, with a shallow 2-layer encoder performing best**. Increasing the depth to three or four layers yields slightly weaker IC and portfolio metrics, which is consistent with over-smoothing and harder optimization under limited structural supervision. This suggests that AlphaSAGE does not require finely tuned encoder depth and is robust to reasonable choices.

**Hidden Dim.** We also probe the effect of the GFlowNet backbone size by varying the hidden dimension on CSI300. Table 7 shows that **a moderate hidden size of 128 achieves the best trade-**

Table 6: Sensitivity to Encoder Layer on CSI300.

| Layers | IC | ICIR | RIC | RICIR | AR | MDD | SR |
|---|---|---|---|---|---|---|---|
| 2 | 0.079 | 0.496 | 0.094 | 0.583 | 7.62% | -17.3% | 1.71 |
| 3 | 0.073 | 0.473 | 0.088 | 0.572 | 7.08% | -18.5% | 1.50 |
| 4 | 0.068 | 0.451 | 0.083 | 0.556 | 6.58% | -19.4% | 1.34 |

**off between expressiveness and generalization**. A smaller size of 64 underfits and degrades both correlation and portfolio metrics, while enlarging to 256 offers no further gains and slightly hurts out-of-sample performance. The qualitative conclusions of the paper remain unchanged across this range, indicating that AlphaSAGE is not overly sensitive to the specific hidden dimension choice.

Table 7: Sensitivity to GFlowNet hidden dimension on CSI300.

| Hidden Dim | IC | ICIR | RIC | RICIR | AR | MDD | SR |
|---|---|---|---|---|---|---|---|
| 64 | 0.059 | 0.368 | 0.078 | 0.464 | 5.42% | -19.6% | 1.31 |
| 128 | 0.079 | 0.496 | 0.094 | 0.583 | 7.62% | -17.3% | 1.71 |
| 256 | 0.073 | 0.448 | 0.089 | 0.561 | 6.88% | -18.1% | 1.59 |

### E.3 REWARD-WEIGHT SCHEDULES

Our multi-dimensional reward combines predictive performance, structure–behaviour alignment, and novelty via time-dependent weights on the structure-aware term $R_{\mathrm{SA}}$ and the novelty term $R_{\mathrm{NOV}}$ (see Eq. (xx)). The design follows a pragmatic curriculum intuition: early in training we encourage exploration and representation learning in the alpha space, while later we place more emphasis on realized predictive performance.

To assess the impact of the weight schedules, we compare three simple choices for the decay function $g(T)$ on CSI300: (i) a constant schedule (no decay), (ii) a linear decay schedule, and (iii) an exponential decay schedule. In all cases, the initial coefficients $(\lambda_0, \eta_0)$ are kept fixed, and only the functional form of $g(T)$ is changed.

Table 8 reports IC/RIC, ICIR/RICIR, and portfolio-level metrics (AR, MDD, SR). Linear decay yields the best overall performance, exponential decay performs similarly, and constant weights underperform. These results indicate that AlphaSAGE is not overly sensitive to the exact schedule shape within a reasonable range, and that annealing the structure-aware and novelty terms is beneficial in practice.

Table 8: Sensitivity to reward-weight schedules on CSI300. Comparison of constant, linear-decay, and exponential-decay schedules for the time-dependent weights on $R_{\mathrm{SA}}$ and $R_{\mathrm{NOV}}$.

| Schedule type | IC | ICIR | RIC | RICIR | AR | MDD | SR |
|---|---|---|---|---|---|---|---|
| Constant | 0.043 | 0.317 | 0.051 | 0.368 | 3.26% | -20.3% | 0.69 |
| Linear decay | 0.079 | 0.496 | 0.094 | 0.583 | 7.62% | -17.3% | 1.71 |
| Exponential decay | 0.075 | 0.482 | 0.095 | 0.578 | 7.56% | -17.6% | 1.69 |

### E.4 ENCODER ARCHITECTURES

Beyond the main results, we also compare a structure aware encoder with a sequence based alternative. In our framework, the default encoder is an RGCN that operates on abstract syntax trees (ASTs) and explicitly exploits the operator feature graph structure. As a reference, we build a Transformer encoder that consumes a linearized expression sequence (for example, prefix notation) under a similar parameter budget.

Table 9 reports the results on CSI300. The RGCN encoder achieves higher correlation metrics (IC, RIC, ICIR, RICIR) and stronger portfolio performance (AR, SR) with a slightly smaller maximum drawdown. This comparison supports the view that the structural inductive bias provided by graph based encoding of ASTs is beneficial for formulaic alpha mining, and that the gains of AlphaSAGE cannot be attributed solely to increased model capacity.

Table 9: Encoder comparison on CSI300. RGCN operates on ASTs, while the Transformer encoder consumes linearized expression sequences under a similar parameter budget.

| Encoder type | IC | ICIR | RIC | RICIR | AR | MDD | SR |
|---|---|---|---|---|---|---|---|
| RGCN | 0.079 | 0.496 | 0.094 | 0.583 | 7.62% | -17.3% | 1.71 |
| Transformer | 0.065 | 0.479 | 0.085 | 0.537 | 4.97% | -18.8% | 1.22 |

### E.5 RL VERSUS GFLOWNET AND THE EFFECT OF STRUCTURE AWARE COMPONENTS

To disentangle the effect of the generative framework from that of the proposed encoder and reward design, we consider four configurations on CSI300: (i) an RL baseline with the original sequential encoding and IC only reward; (ii) the same RL baseline augmented with the RGCN encoder and the composite reward; (iii) a GFlowNet with the original sequential encoding and IC only reward; and (iv) a GFlowNet with the RGCN encoder and the composite reward, which corresponds to AlphaSAGE.

Table 10 summarizes the results. Two observations are consistent with the main text. First, adding the structure aware components improves both RL and GFlowNet, increasing IC/RIC as well as AR and SR while reducing maximum drawdown. Second, without these components the GFlowNet configuration struggles under sparse IC only feedback and underperforms the RL baseline, whereas with the RGCN encoder and composite reward it achieves the best overall performance. This supports our claim that both the choice of GFlowNet and the structure aware design are important, and that the gains of AlphaSAGE come from their combination.

Table 10: RL versus GFlowNet with and without structure aware components on CSI300. "Proposed components" denotes the use of the RGCN encoder together with the composite reward.

| Framework | Components | IC | ICIR | RIC | RICIR | AR | MDD | SR |
|---|---|---|---|---|---|---|---|---|
| RL | ✗ | 0.058 | 0.414 | 0.057 | 0.360 | 4.00% | -22.6% | 0.76 |
| RL | ✓ | 0.067 | 0.459 | 0.079 | 0.502 | 5.53% | -20.7% | 1.03 |
| GFlowNet | ✗ | 0.046 | 0.313 | 0.060 | 0.397 | -0.47% | -24.8% | -0.11 |
| GFlowNet | ✓ | **0.079** | **0.496** | **0.094** | **0.583** | **7.62%** | **-17.3%** | **1.71** |

### E.6 RUNTIME AND SCALABILITY

In our implementation, the dominant computational cost of AlphaSAGE comes from evaluating candidate alphas on historical cross-sectional panels. This cost is shared with existing learning-based alpha mining systems such as AlphaGen, AlphaQCM, and AlphaForge, since all of them must compute panel-wise values for large collections of formulaic alphas. The additional components introduced by AlphaSAGE are relatively lightweight.

First, the GFlowNet sampler operates on partial abstract syntax trees (ASTs) with a bounded maximum length `MaxLen`. The number of sampling steps per alpha is therefore proportional to `MaxLen` and comparable in magnitude to the sequential expression-generation steps used in RL-based frameworks. Second, the structure-aware reward only requires computing $k$-nearest neighbors in a low-dimensional graph-embedding space. These embeddings and neighbor relationships are precomputed and cached during training, so the incremental overhead is modest. Third, the early-stopping mechanism controlled by `MaxLen` stochastically truncates trajectories once a valid expression of sufficient length is formed, which avoids wasteful exploration of overly long expression trees and empirically accelerates convergence.

In typical quantitative trading workflows, alpha mining is run offline at a relatively low frequency (for example weekly or monthly) to update the alpha library. During portfolio construction and execution, the system evaluates only the fixed set of discovered alphas, so the latency of the mining model does not directly affect real-time trading decisions.

Table 11 reports training time for the main methods on several universes. AlphaSAGE is efficient in practice and suitable for large-scale alpha mining under realistic resource constraints.

Table 11: Runtime comparison (hours) on different universes.

| Method | CSI300 | CSI500 | S&P 500 | CSI1000 |
|---|---|---|---|---|
| AlphaGen | 1.34 | 1.68 | 1.48 | 2.11 |
| AlphaQCM | 1.72 | 1.98 | 1.82 | 2.32 |
| AlphaForge | 1.56 | 2.16 | 2.02 | 2.74 |
| AlphaSAGE | **0.49** | **0.76** | **0.71** | **1.12** |

### E.7 STABILITY ACROSS RANDOM SEEDS

To assess robustness with respect to random initialization, we ran multiple independent trials on CSI300 for all RL-based methods using 5 different random seeds and report mean and standard deviation for each metric. As shown in Table 12, AlphaSAGE consistently outperforms the baselines across correlation metrics (IC/RIC, ICIR/RICIR) and portfolio metrics (AR, SR) while maintaining a smaller maximum drawdown. The ranking between methods is stable across runs, indicating that our main conclusions are not driven by a particular seed.

Table 12: Multiple runs on CSI300. Mean (standard deviation) over different random seeds.

| Method | IC | ICIR | RIC | RICIR | AR | MDD | SR |
|---|---|---|---|---|---|---|---|
| AlphaGen | 0.063 | 0.454 | 0.061 | 0.389 | 4.18% | -22.3% | 0.84 |
| | (0.005) | (0.024) | (0.006) | (0.021) | (0.004) | (0.017) | (0.126) |
| AlphaQCM | 0.045 | 0.268 | 0.044 | 0.259 | 2.23% | -23.8% | 0.52 |
| | (0.002) | (0.015) | (0.006) | (0.019) | (0.003) | (0.019) | (0.155) |
| AlphaForge | 0.040 | 0.246 | 0.049 | 0.291 | 4.01% | -21.3% | 0.96 |
| | (0.003) | (0.017) | (0.005) | (0.022) | (0.003) | (0.020) | (0.112) |
| AlphaSAGE | 0.073 | 0.473 | 0.088 | 0.572 | 7.08% | -18.5% | 1.50 |
| | (0.004) | (0.022) | (0.008) | (0.026) | (0.005) | (0.022) | (0.143) |

### E.8 SUBPERIOD PERFORMANCE UNDER REGIME SHIFTS

In the Appendix E.1 we report that AlphaSAGE delivers consistently strong IC- and ICIR-based performance across all markets and horizons, while cumulative return curves on CSI500 (2022–2024) and S&P500 (2018–2020) are less pronounced and in some cases slightly below certain baselines. This section provides additional context for these findings.

First, AlphaSAGE is trained to discover alphas by maximizing a reward in which $R_{\text{IC}}$ is the primary component. Information coefficient measures the average cross-sectional correlation between an alpha signal and next-period returns and is a useful proxy for ranking quality, but it does not directly optimize path-dependent portfolio outcomes. Over short or turbulent subperiods, an alpha library with stable long-horizon positive IC can still produce flatter cumulative returns if the window exhibits unusually high volatility, sharp factor-premia reversals, elevated turnover costs, or compressed cross-sectional dispersion. In such regimes the mapping from cross-sectional predictability to realized P&L can weaken temporarily, so IC-oriented discovery does not guarantee dominance in every short subwindow.

Second, the CSI500 interval from 2022 to 2024 coincides with several well-documented regime events that disproportionately affected mid- and small-cap names and induced repeated style rota-

tions. In 2022, strict zero-COVID policies and the Shanghai lockdown weighed heavily on economic activity and risk appetite; the ongoing property-sector stress and mortgage-related incidents further pushed the market into a risk-off stance. After the late-2022 policy pivot and reopening, equity leadership rotated quickly across sectors, and in 2023–2024 policy support increasingly emphasized high-dividend and buyback themes that favored larger, more stable firms over the CSI500 universe. Against this backdrop, an IC-optimized alpha library can exhibit relatively muted cumulative returns on CSI500 over this specific window even if its average predictive edge remains positive.

Third, the S&P500 window from 2018 to 2020 likewise spans several abrupt regime shifts. The escalation of the U.S.–China trade conflict and tighter monetary policy in 2018 contributed to a sharp Q4 sell-off and rapid factor rotations; in 2019, persistent trade uncertainty and growth concerns continued to destabilize style leadership. In early 2020, the COVID-19 shock produced an unprecedented crash followed by a policy-driven rebound with extreme cross-sectional dispersion. Such environments are known to distort the short-horizon relationship between historical IC and realized portfolio performance.

Overall, these subperiods illustrate that AlphaSAGE's objective—maximizing an IC-dominated reward over long histories—targets robust cross-sectional predictability, while realized cumulative returns in short, highly stressed windows can be shaped by regime-specific shocks and style dynamics that lie beyond the scope of the discovery objective.

Table 13: Performance Comparison of Different Methods on CSI300, CSI500, CSI1000 (China) and S&P500 (U.S.). Bold and underlined numbers represent the best and second-best performance across all compared approaches, respectively.

| Method | *IC* | *RankIC* | *AR* | *MDD* |
|---|---|---|---|---|
| MLP | 0.020 | 0.019 | 3.54% | -20.9% |
| LightGBM | 0.011 | 0.006 | 2.61% | -18.5% |
| XGBoost | 0.031 | 0.033 | 5.40% | -17.5% |
| GP | 0.026 | 0.028 | 6.80% | -17.6% |
| AlphaGen | 0.058 | 0.057 | 4.00% | -22.6% |
| AlphaQCM | 0.043 | 0.042 | 1.95% | -24.8% |
| AlphaForge | 0.041 | 0.052 | 3.90% | -21.9% |
| AlphaAgent | 0.051 | 0.056 | 2.16% | -26.9% |
| **AlphaSAGE(ours)** | **0.079** | **0.094** | **7.62%** | **-17.3%** |

## F PROOF

**Proposition F.1** (Alpha Diversity Stabilizes Estimation and Prediction). *Let $F = [\alpha_1, \ldots, \alpha_N] \in \mathbb{R}^{T \times N}$ collect $N$ standardized alphas (each column has mean $0$ and variance $1$), and let $y \in \mathbb{R}^T$ be the target return. Define*

$$\Sigma = \frac{1}{T}F^\top F \in \mathbb{R}^{N \times N}, \qquad g = \frac{1}{T}F^\top y \in \mathbb{R}^N. \tag{35}$$

*The OLS estimator is $\hat{\beta} = \Sigma^{-1}g$. Suppose $\epsilon = y - F\beta^\star$ satisfies $\mathbb{E}[\epsilon] = 0$ and $\mathrm{Var}(\epsilon) = \sigma^2 I_T$. Then:*

1. *(Estimator variance decomposition) Writing the eigendecomposition $\Sigma = Q\Lambda Q^\top$ with eigenvalues $1 \geq \lambda_1 \geq \cdots \geq \lambda_N > 0$, we have*

$$\mathrm{Var}(\hat{\beta}) = \frac{\sigma^2}{T}\Sigma^{-1} \quad \Rightarrow \quad \mathrm{tr}\,\mathrm{Var}(\hat{\beta}) = \frac{\sigma^2}{T}\sum_{i=1}^{N}\frac{1}{\lambda_i}. \tag{36}$$

*Consequently, as pairwise correlations increase and the spectrum becomes more ill-conditioned (small $\lambda_{\min}$), the total estimation variance inflates.*

2. *(Prediction risk amplification) The in-sample prediction variance satisfies*

$$\mathbb{E}\big[\|F(\hat{\beta}-\beta^{\star})\|_2^2\big] = \text{tr}\big(F \text{ Var}(\hat{\beta}) F^{\top}\big) = \frac{\sigma^2}{T} \text{tr}\big(F\Sigma^{-1}F^{\top}\big) = \sigma^2 \text{ tr}\big(\Sigma \Sigma^{-1}\big) = \sigma^2 N, \tag{37}$$

*but the* out-of-sample *risk for a new design with the same second moments equals*

$$\mathcal{R}_{pred} = \sigma^2 + \mathbb{E}\big[(\alpha^{\top}(\hat{\beta}-\beta^{\star}))^2\big] = \sigma^2 + \frac{\sigma^2}{T} \text{tr}(\Sigma \Sigma^{-1}) = \sigma^2\Big(1 + \frac{N}{T}\Big), \tag{38}$$

*while the* uncertainty allocation across coordinates *is governed by* $\Sigma^{-1}$: *higher multi-collinearity (smaller* $\lambda_{\min}$) *yields larger coordinate-wise dispersion of* $\hat{\beta}$ *and hence less interpretability.*

3. *(Sensitivity to perturbations) For perturbations* $(\Delta\Sigma, \Delta g)$, *the linear system* $\Sigma\hat{\beta} = g$ *obeys the classical bound*

$$\frac{\|\Delta\hat{\beta}\|_2}{\|\hat{\beta}\|_2} \lesssim \kappa_2(\Sigma)\Big(\frac{\|\Delta g\|_2}{\|g\|_2} + \frac{\|\Delta\Sigma\|_2}{\|\Sigma\|_2}\Big), \tag{39}$$

*where* $\kappa_2(\Sigma) = \|\Sigma\|_2 \|\Sigma^{-1}\|_2 = \lambda_{\max}/\lambda_{\min}$. *Thus, near-collinearity (large* $\kappa_2$) *makes* $\hat{\beta}$ *highly unstable under small data noise or distributional drift.*

4. *(Two-alpha closed form) For two standardized alphas with correlation* $\rho$,

$$\Sigma = \begin{bmatrix} 1 & \rho \\ \rho & 1 \end{bmatrix}, \qquad \Sigma^{-1} = \frac{1}{1-\rho^2}\begin{bmatrix} 1 & -\rho \\ -\rho & 1 \end{bmatrix}, \tag{40}$$

*so* $\text{Var}(\hat{\beta}) = \frac{\sigma^2}{T(1-\rho^2)}\begin{bmatrix} 1 & -\rho \\ -\rho & 1 \end{bmatrix}$ *and* $\kappa_2(\Sigma) = \frac{1+\rho}{1-\rho}$. *As* $\rho \to 1$, *both the variance and the condition number blow up.*

5. *(Equicorrelated family) If* $\Sigma$ *has equicorrelation* $\rho$ *off-diagonal, then*

$$\lambda_1 = 1 + (N-1)\rho, \qquad \lambda_2 = \cdots = \lambda_N = 1 - \rho, \tag{41}$$

*and hence*

$$\text{tr Var}(\hat{\beta}) = \frac{\sigma^2}{T}\Big(\frac{1}{1+(N-1)\rho} + \frac{N-1}{1-\rho}\Big), \qquad \kappa_2(\Sigma) = \frac{1+(N-1)\rho}{1-\rho}. \tag{42}$$

*Even modest* $\rho > 0$ *causes variance inflation linear in* $N$ *through the* $(N-1)/(1-\rho)$ *term; promoting diversity (smaller* $\rho$) *sharply reduces this inflation.*

*Proof.* (1) Since $\hat{\beta} = (F^{\top}F)^{-1}F^{\top}y = \Sigma^{-1}g$ and $y = F\beta^{\star} + \epsilon$ with $\text{Var}(\epsilon) = \sigma^2 I_T$, we have

$$\text{Var}(\hat{\beta}) = \Sigma^{-1}\Big(\frac{1}{T^2}F^{\top} \text{ Var}(y) F\Big)\Sigma^{-1} = \Sigma^{-1}\Big(\frac{\sigma^2}{T^2}F^{\top}F\Big)\Sigma^{-1} = \frac{\sigma^2}{T}\Sigma^{-1}. \tag{43}$$

Using $\Sigma = Q\Lambda Q^{\top}$ yields $\text{tr Var}(\hat{\beta}) = \frac{\sigma^2}{T}\sum_i \lambda_i^{-1}$.

(2) For in-sample variance,

$$\mathbb{E}\|F(\hat{\beta}-\beta^{\star})\|_2^2 = \text{tr}\big(F \text{ Var}(\hat{\beta}) F^{\top}\big) = \frac{\sigma^2}{T} \text{tr}(F\Sigma^{-1}F^{\top}). \tag{44}$$

Since $F\Sigma^{-1}F^{\top}$ and $\Sigma\Sigma^{-1}$ share the same trace ($\text{tr}(AB) = \text{tr}(BA)$), this equals $\sigma^2 \text{ tr}(I_N) = \sigma^2 N$. For a new draw $\tilde{\alpha}$ with the same second moments, $\mathbb{E}[\tilde{\alpha}\tilde{\alpha}^{\top}] = \Sigma$, so the added generalization variance is $\frac{\sigma^2}{T} \text{tr}(\Sigma\Sigma^{-1}) = \frac{\sigma^2 N}{T}$, giving $\mathcal{R}_{\text{pred}} = \sigma^2(1 + N/T)$; however the *distribution* of this

uncertainty over coordinates is governed by $\Sigma^{-1}$, worsening with ill-conditioning, which harms interpretability of individual $\hat{\beta}_i$.

(3) The stated perturbation bound follows from standard linear system sensitivity: for $\Sigma\hat{\beta} = g$, first-order analysis (or the Bauer–Fike–type arguments) gives $\|\Delta\hat{\beta}\|_2 \lesssim \|\Sigma^{-1}\|_2\big(\|\Delta g\|_2 + \|\Delta\Sigma\|_2\|\hat{\beta}\|_2\big)$; normalizing by $\|\hat{\beta}\|_2$ and noting $\|\Sigma^{-1}\|_2\,\|\Sigma\|_2 = \kappa_2(\Sigma)$ yields the claim.

(4)–(5) The two-alpha and equicorrelation calculations follow from direct inversion and the known eigenstructure: for equicorrelation, the all-ones vector is the top eigenvector with eigenvalue $1 + (N-1)\rho$ and the orthogonal complement has eigenvalue $1 - \rho$. Plugging these into part (1) gives the trace formula and condition number.

Collectively, (1)–(5) show that reducing off-diagonal correlations increases the eigenvalues of $\Sigma$, decreases $\kappa_2(\Sigma)$, shrinks $\mathrm{Var}(\hat{\beta})$, and improves stability and interpretability—formally substantiating the need for diverse, weakly correlated alphas. $\qquad\square$

**Corollary F.2** (Regularization as a proxy for diversity). *For ridge with penalty $\lambda > 0$, $\hat{\beta}_\lambda = (\Sigma + \lambda I)^{-1}g$ and*

$$\mathrm{Var}(\hat{\beta}_\lambda) = \frac{\sigma^2}{T}(\Sigma + \lambda I)^{-1} \quad \Rightarrow \quad \mathrm{tr}\,\mathrm{Var}(\hat{\beta}_\lambda) = \frac{\sigma^2}{T}\sum_{i=1}^{N}\frac{1}{\lambda_i + \lambda}. \tag{45}$$

*Either increasing diversity (raising the $\lambda_i$) or increasing $\lambda$ reduces variance; explicit diversity control targets the spectrum directly, often achieving lower variance without the shrinkage bias inherent in ridge.*

