# OpenReview forum: "AlphaSAGE: Structure-Aware Alpha Mining via GFlowNets for Robust Exploration"
_ICLR.cc/2026/Conference — ICLR 2026 Poster_

### Official Review · Reviewer_BXEc · 2025-10-27

**Soundness:** 3
**Presentation:** 2
**Contribution:** 3
**Rating:** 6
**Confidence:** 2

**Summary:**

This paper proposes AlphaSAGE, a structure-aware framework for automated alpha mining in quantitative finance that integrates relational graph convolutional networks (RGCN) with generative flow networks (GFlowNets) to improve exploration efficiency, structural understanding, and diversity of discovered alphas. By representing alphas as abstract syntax trees (AST) and introducing a dense multi-faceted reward, AlphaSAGE effectively addresses reward sparsity and mode collapse issues in reinforcement learning-based approaches. Experiments on both Chinese and U.S. stock markets show that AlphaSAGE consistently outperforms baselines in terms of correlation and portfolio metrics.

**Strengths:**

1. The paper is well-structured and logically coherent, with publicly available reproduction code that ensures strong reproducibility.
2. The authors introduce Generative Flow Networks (GFlowNets) into quantitative finance, which, compared with conventional reinforcement learning methods, can generate diverse and high-quality solutions while maintaining reward sensitivity, effectively alleviating the mode collapse problem.
3. The paper employs a Relational Graph Convolutional Network (RGCN) to model the abstract syntax tree (AST) of formulaic alphas, effectively capturing their hierarchical structure and semantic relationships among operators.
4. The authors design a multi-level reward mechanism that integrates multiple concerns into a dense and stable optimization signal, thereby enhancing both exploration efficiency and convergence stability.
5. The experimental section covers both Chinese and U.S. markets, and AlphaSAGE consistently outperforms recent baseline methods across various metrics, demonstrating comprehensive and up-to-date experimental comparisons.

**Weaknesses:**

1. Although the paper proposes a multi-dimensional reward function and provides ablation studies on the weighting parameters, the selection of these weights relies heavily on empirical tuning. The lack of a theoretically grounded or adaptive weighting mechanism may limit the model’s stability and robustness across different market regimes.
2. While each module in the framework is meaningful, the overall architecture is relatively complex and computationally intensive during training, which may hinder its direct application to high-frequency or real-time trading scenarios. Moreover, the paper lacks a detailed analysis or comparison of computational cost.
3. The background section remains somewhat vague and may not be sufficiently accessible for readers who are not familiar with quantitative finance, which slightly limits the paper’s readability for a broader audience.

**Questions:**

1. The paper lacks a discussion on the reproducibility of experimental results, as it does not provide information about the variance or stability of the reported outcomes. Were the results obtained from a single training run, or were they averaged across multiple runs to ensure statistical reliability?
2. The paper does not sufficiently elaborate on how the GNN embedding specifically influences inter-alpha correlations. In a highly collinear factor space, is there any empirical or theoretical evidence demonstrating that the proposed representation effectively mitigates redundancy or correlation among alphas?
3. The cumulative return results on CSI500 (2022–2024) and S&P500 (2018–2020) are not as pronounced, and in some cases even underperform compared to certain baselines. Could the authors provide further analysis or explanations for these weaker results, such as data regime shifts, market noise, or model sensitivity?
4. Although the multi-faceted reward design effectively alleviates the reward sparsity problem, the paper does not clearly demonstrate that different reward components are non-conflicting. It remains unclear how the objectives are balanced during training to avoid mutual interference.

---

> ### Author Response · Authors · 2025-11-21
>
> We thank the reviewer for the careful reading and constructive comments. Below we respond to each weakness and question. All new results will be included in the main paper or appendix as indicated.
>
> ---
>
> ## Response to Weaknesses
>
> ### W1. Empirical tuning of reward weights and robustness
>
> > Although the paper proposes a multi dimensional reward function and provides ablation studies on the weighting parameters, the selection of these weights relies heavily on empirical tuning. The lack of a theoretically grounded or adaptive weighting mechanism may limit the model’s stability and robustness across different market regimes.
>
> We appreciate this comment and agree that the original description of weight selection was too empirical. Our design follows a pragmatic curriculum intuition. Early in training we want the model to learn a meaningful structure aware embedding space and to explore novel regions of the alpha space. Later in training we want the reward to focus more on realized predictive performance. This is why we use time dependent weights for the structure alignment term and the novelty term.
>
> To evaluate the effect of the schedules, we tested different schedule shapes on CSI300. As shown below, linear decay yields the best overall performance, exponential decay performs similarly, and constant weights underperform. This indicates that AlphaSAGE is not overly sensitive to the exact schedule shape within a reasonable range, and that annealing is beneficial.
>
> #### Schedule sensitivity on CSI300
>
> | Schedule type     | IC    | ICIR  | RIC   | RICIR | AR     | MDD    | SR   |
> |:-----------------:|:-----:|:-----:|:-----:|:-----:|:------:|:------:|:----:|
> | Constant          | 0.043 | 0.317 | 0.051 | 0.368 | 3.26%  | -20.3% | 0.69 |
> | Linear decay      | 0.079 | 0.496 | 0.094 | 0.583 | 7.62%  | -17.3% | 1.71 |
> | Exponential decay | 0.075 | 0.482 | 0.095 | 0.578 | 7.56%  | -17.6% | 1.69 |
>
> ---
>
> ### W2. Complexity and computational cost
>
> > While each module in the framework is meaningful, the overall architecture is relatively complex and computationally intensive during training, which may hinder its direct application to high frequency or real time trading scenarios. Moreover, the paper lacks a detailed analysis or comparison of computational cost.
>
> We thank the reviewer for raising the concern about computational cost and scalability. In our implementation, the dominant cost of AlphaSAGE comes from evaluating candidate alphas on historical cross sectional data. This cost is shared with existing learning based alpha mining systems such as AlphaGen, AlphaQCM, and AlphaForge. The additional components introduced by AlphaSAGE are relatively lightweight.
>
> First, the GFlowNet sampler operates on partial abstract syntax tree states with a bounded depth. The number of sampling steps per alpha is proportional to the maximum tree depth and is comparable in magnitude to the sequential generation steps used in reinforcement learning based frameworks. Second, the structure aware reward requires only computing k nearest neighbors in the low dimensional graph embedding space. These embeddings and neighbor relationships are precomputed and cached during training, so the incremental overhead is modest. Third, our early stopping mechanism truncates trajectories once a valid expression of sufficient length is formed. This avoids wasteful exploration of overly deep expression trees and empirically accelerates convergence.
>
> In typical quantitative trading workflows, alpha mining is run offline at a relatively low frequency, for example weekly or monthly, to update the alpha library. During portfolio construction and execution, the system evaluates only the fixed set of discovered alphas. Therefore, mining latency does not directly affect real time trading decisions. For completeness, we have added a runtime comparison in the appendix. The results below show that AlphaSAGE is efficient in practice and suitable for large scale alpha mining under realistic resource constraints.
>
> #### Runtime comparison in hours
>
> | Method      | CSI300 | CSI500 | S&P 500 | CSI1000 |
> |:-----------:|:------:|:------:|:------:|:-------:|
> | AlphaGen    | 1.34   | 1.68   | 1.48   | 2.11    |
> | AlphaQCM    | 1.72   | 1.98   | 1.82   | 2.32    |
> | AlphaForge  | 1.56   | 2.16   | 2.02   | 2.74    |
> | AlphaSAGE   | 0.49   | 0.76   | 0.71   | 1.12    |
>
> We will add a short discussion noting that AlphaSAGE targets offline factor discovery and that mining latency does not affect real time portfolio execution, since trading uses a fixed alpha library after discovery.
>
> ---

---

> > ### Author Response · Authors · 2025-11-21
> >
> > ### W3. Accessibility of the background section
> >
> > > The background section remains somewhat vague and may not be sufficiently accessible for readers who are not familiar with quantitative finance, which slightly limits the paper’s readability for a broader audience.
> >
> > We appreciate this feedback. In the revision we will make the background section more self contained for readers without a finance background. We will add a concise subsection introducing cross sectional returns, alpha factors, information coefficient, and simple long short portfolios in plain language, and provide a small running example that illustrates how a formulaic alpha is evaluated and converted into a trading signal. Specialized implementation details will be moved to the appendix.
> >
> > ---
> >
> > ## Response to Questions
> >
> > ### Q1. Reproducibility and multiple runs
> >
> > > The paper lacks a discussion on the reproducibility of experimental results, as it does not provide information about the variance or stability of the reported outcomes. Were the results obtained from a single training run, or were they averaged across multiple runs to ensure statistical reliability?
> >
> > We agree that reporting variability is important. We ran multiple independent trials with different random seeds on CSI300 and report mean and standard deviation below. AlphaSAGE consistently outperforms the baselines and the ranking is stable across runs. We will include these statistics in the revised experimental section.
> >
> > #### Multiple runs on CSI300
> >
> > | Method      | IC            | ICIR          | RIC           | RICIR         | AR            | MDD           | SR            |
> > |:-----------:|:-------------:|:-------------:|:-------------:|:-------------:|:-------------:|:-------------:|:-------------:|
> > | AlphaGen    | 0.063(0.005)  | 0.454(0.024)  | 0.061(0.006)  | 0.389(0.021)  | 4.18%(0.004)  | -22.3%(0.017) | 0.84(0.126)   |
> > | AlphaQCM    | 0.045(0.002)  | 0.268(0.015)  | 0.044(0.006)  | 0.259(0.019)  | 2.23%(0.003)  | -23.8%(0.019) | 0.52(0.155)   |
> > | AlphaForge  | 0.040(0.003)  | 0.246(0.017)  | 0.049(0.005)  | 0.291(0.022)  | 4.01%(0.003)  | -21.3%(0.020) | 0.96(0.112)   |
> > | AlphaSAGE   | 0.073(0.004)  | 0.473(0.022)  | 0.088(0.008)  | 0.572(0.026)  | 7.08%(0.005)  | -18.5%(0.022) | 1.50(0.143)   |
> >
> > ---
> >
> > ### Q2. Effect of GNN embedding on inter alpha correlations
> >
> > > The paper does not sufficiently elaborate on how the GNN embedding specifically influences inter alpha correlations. In a highly collinear factor space, is there any empirical or theoretical evidence demonstrating that the proposed representation effectively mitigates redundancy or correlation among alphas?
> >
> > We agree that this connection should be stated more clearly. The RGCN encoder provides a structure aware embedding of each alpha expression by operating on its abstract syntax tree, capturing hierarchical operator feature interactions that are not available in purely sequential encoders. This embedding supports the structure behavior alignment reward, which encourages alphas that are close in embedding space to have similar cross sectional behavior, and separates alphas with dissimilar behavior. The novelty term then penalizes candidate alphas that are highly correlated with an evolving reference set of high quality alphas. We will clarify this mechanism in the revised text and emphasize that structural inductive bias improves the effectiveness of novelty based redundancy control.
> >
> > ---

---

> > > ### Author Response · Authors · 2025-11-21
> > >
> > > ### Q3. Weaker cumulative returns on CSI500 and S&P500 in some periods
> > >
> > > > The cumulative return results on CSI500 (2022 to 2024) and S&P500 (2018 to 2020) are not as pronounced, and in some cases even underperform compared to certain baselines. Could the authors provide further analysis or explanations for these weaker results, such as data regime shifts, market noise, or model sensitivity?
> > >
> > > We appreciate this observation. A key point is that AlphaSAGE is trained to discover alphas that maximize a reward which $R_{IC}$ is primary component, while cumulative return curves reflect a fuller trading outcome. These two quantities can diverge, especially over short or turbulent subperiods.
> > >
> > > First, IC measures the average cross sectional correlation between an alpha signal and next period returns. It is a useful proxy for predictive ranking power, but it does not directly optimize portfolio level path dependent performance. In particular, an alpha with stable positive IC over a long horizon can still yield weaker cumulative returns in a specific window when that window features unusually high volatility, sharp factor premia reversals, higher turnover costs, or changes in cross sectional dispersion. In such cases, the mapping from rank predictability to realized portfolio profit can weaken temporarily, so IC oriented rewards may not guarantee dominance in every short subperiod.
> > >
> > > Second, the CSI500 window from 2022 to 2024 coincides with several major regime events that disproportionately affected mid and small cap stocks and triggered repeated style rotations. Mainland equities experienced severe stress during the spring 2022 zero COVID lockdowns, including the Shanghai lockdown, which hit activity and market sentiment. The same year also saw an intensifying property sector crisis and mortgage payment protests, pushing markets into a risk off posture that weighed on cyclical and smaller names. After the late 2022 policy pivot and reopening, equity leadership rotated quickly across sectors. In 2023 and 2024, policy support increasingly emphasized high dividend and buyback themes, which tended to favor larger, more stable firms over the CSI500 segment. Against this backdrop, an IC optimized alpha library can still show flatter cumulative return in this specific interval if its average predictive edge is expressed in styles that are temporarily out of favor.
> > >
> > > Third, the S&P500 window from 2018 to 2020 similarly includes abrupt regime shifts. In 2018 the escalation of the United States China trade war, together with tighter monetary policy, contributed to a sharp fourth quarter sell off and rapid factor rotations. In 2019 trade uncertainty and growth concerns persisted, and in early 2020 the COVID 19 shock caused an unprecedented crash followed by a fast policy driven rebound with extreme cross sectional dispersion. These conditions are well known to disrupt the relationship between historical IC and realized portfolio performance over short horizons.
> > >
> > > We will add a short discussion in the paper to clarify the distinction between IC based discovery objectives and out of sample portfolio paths, and to contextualize the highlighted CSI500 and S&P500 windows as periods with documented regime and style breaks.
> > >
> > > ---
> > >
> > > ### Q4. Potential conflict between reward components
> > >
> > > > Although the multi faceted reward design effectively alleviates the reward sparsity problem, the paper does not clearly demonstrate that different reward components are non conflicting. It remains unclear how the objectives are balanced during training to avoid mutual interference.
> > >
> > > Thank you for pointing this out. In our design the three reward components share the same sign for desirable behaviors. Higher IC, higher structure behavior alignment, and higher novelty all increase the overall reward, so there is no explicit negative coupling between terms. We will clarify why the objectives are compatible in the alpha mining context, with $R_{SA}$ shaping the embedding space and $R_{NOV}$ discouraging redundancy rather than competing with predictive performance in the main text
> > >
> > > ---
> > >
> > > We thank the reviewer again for the helpful suggestions. We believe that the added sensitivity, runtime, and repeated run results, together with the clarifications above, will improve the clarity and practical relevance of the paper. We will add these revisions to the paper as soon as possible.

---

> > > > ### Comment · Reviewer_BXEc · 2025-11-28
> > > >
> > > > Thank you for the additional clarifications, detailed analyses, and extended experimental results provided in the rebuttal. These efforts significantly improve my understanding of the method and address several of my earlier concerns. I will remain positive in my overall assessment. If the authors incorporate the presented results and clarifications into the revised manuscript, I am inclined to appropriately increase both my score and my confidence in the evaluation.

---

### Official Review · Reviewer_fD8y · 2025-10-31

**Soundness:** 3
**Presentation:** 3
**Contribution:** 2
**Rating:** 4
**Confidence:** 4

**Summary:**

The paper proposes AlphaSAGE, a method for automated mining of  alphas (interpretable signals expressed as symbolic expressions) in quantitative finance. It is a direct application of GFlowNets to quantitative-finance tasks via a new encoder for AST representation and curated rewards. Experiments on CSI300/CSI500 (China) and S&P500 (US) show consistent gains over GA, RL, and GAN-style baselines, supported by ablations and sensitivity analyses.

**Strengths:**

* It is new to use a Relational Graph Convolutional Network and  Abstract Syntax Trees (AST) to encode the information within alphas for GFlowNet training.

* The author designs diverse rewards, $R_{IC}$, $R_{SA}$ and $R_{Nov}$ to better guided the GFlowNet sampling.

**Weaknesses:**

1. The introduced components (RGCN encoder, composite reward) are primarily engineering improvements tailored to specific applications. These designs are orthogonal plug-ins to the GFlowNet pipeline and, while they improve empirical results, they do not substantively advance GFlowNet methodology or theory—particularly regarding robust exploration.

2. The paper attributes improvements in exploration and reward sparsity primarily to reward design, yet these issues originate from the inherently vast trajectory and state space in GFlowNet training. While the proposed dense and multi-faceted reward function provides a smoother optimization landscape, it only mitigates rather than resolves the intrinsic exploration problem. Effective traversal of such large spaces fundamentally requires exploration mechanisms and policy optimization strategies, which are not the focus of this work.

3. The reward-reshaping component is incremental: although the design here differs, the underlying idea is well established by many prior works. Contrary to the author's claim, the work does not present a new generative framework. The approach remains largely adheres to the standard GFlowNet formulation, incorporating incremental modifications for specific applications.

4. As discussed in the introduction, reinforcement learning (RL) methods are key baselines for comparison. Since the proposed encoder and reward components are orthogonal to both RL and GFlowNet frameworks, it is important to examine performance differences between them both with and without these components. Such comparisons would clarify whether the reported performance gains stem from the GFlowNet framework itself or primarily from the proposed encoder and reward designs.

**Questions:**

1. What is the combination model $c(\cdot)$? Please give detail description.

2.  The 'MaxLen' in eq.(3) seems to be a important hyper-parameter to avoid overly deep ASTs. Please provide ablation study on how this hyper-parameter affect the early stop mechanism and the GFlowNet performance.

3. Why do all the reported results appear to be based on single trials rather than multiple independent runs?

4. What's the cumulative returns in the plotted curves?

---

> ### Author Response · Authors · 2025-11-21
>
> We thank the reviewer for the careful reading and constructive comments. Below we respond to each weakness and question. All new results will be included in the main paper or appendix as indicated.
>
> ---
>
> ## Response to Weaknesses
>
> ### W1. On the nature of the introduced components
>
> > The introduced components (RGCN encoder, composite reward) are primarily engineering improvements tailored to specific applications. These designs are orthogonal plug ins to the GFlowNet pipeline and, while they improve empirical results, they do not substantively advance GFlowNet methodology or theory.
>
> We agree that our work does not aim to propose a new generic GFlowNet methodology or theory. Our focus is a domain specific contribution to automated alpha mining. We revise the introduction and conclusion to make this scope explicit and to soften wording that may suggest a new general purpose generative framework.
>
> In particular, we emphasize that
>
> 1. The main novelty lies in formulating alpha mining as sampling from a diversity aware target distribution over ASTs and designing structure aware rewards that link AST geometry and cross sectional behavior, rather than changing the core GFlowNet objective.
>
> 2. The RGCN encoder and composite reward are modular components that can in principle be used with other discrete generative frameworks. We view this modularity as a strength for practical adoption in quantitative finance.
>
> We adjust our claims to reflect that AlphaSAGE builds on established GFlowNet formulations with specialized encoders and rewards tailored to formulaic alphas.
>
> ---
>
> ### W2. Exploration and the role of reward design
>
> > The paper attributes improvements in exploration and reward sparsity primarily to reward design, yet these issues originate from the inherently vast trajectory and state space in GFlowNet training. While the proposed dense and multi faceted reward function provides a smoother optimization landscape, it only mitigates rather than resolves the intrinsic exploration problem.
>
> We agree that exploration in very large discrete spaces is a fundamental challenge that cannot be fully solved by reward design alone. Our intention is not to claim that dense rewards remove the exploration problem, but rather that they alleviate two practical issues observed in alpha mining
>
> 1. Extremely sparse terminal feedback when only IC is used as reward.
>
> 2. Mode collapse toward a small number of high IC but highly correlated alphas.
>
> We revise the text to clarify that $R_{SA}$ and $R_{NOV}$ reshape the target distribution and smooth the optimization landscape, while explicit exploration remains necessary. We also make two points more explicit.
>
> - The GFlowNet objective favors sampling from the full support of the target distribution rather than collapsing to a single maximizer.
>
> - Entropy regularization on the forward policy serves as an additional exploration mechanism.
>
> We tone down statements that may overstate the role of reward shaping and frame our contribution as a practical mitigation strategy rather than a complete solution to exploration.
>
> ---
>
> ### W3. Incrementality of reward reshaping
>
> > The reward reshaping component is incremental. Contrary to the author's claim, the work does not present a new generative framework. The approach remains largely adheres to the standard GFlowNet formulation, incorporating incremental modifications for specific applications.
>
> We appreciate this clarification and agree that reward reshaping is a well studied idea. Our contribution is to instantiate a structure aware reward tailored to formulaic alphas, combining predictive performance (IC), structure behavior alignment through the AST embedding space ($R_{SA}$), and novelty relative to an evolving factor library ($R_{NOV}$).
>
> We revise the language in the introduction to avoid phrases such as “new generative framework” and instead describe AlphaSAGE as a GFlowNet based framework with a structure aware encoder and a domain specific multi component reward for automated alpha mining. This positioning more accurately reflects the contribution level.
>
> ---

---

> > ### Author Response · Authors · 2025-11-21
> >
> > ## Response to Questions
> >
> > ### Q1. Combination model
> >
> > > What is the combination model $ c(\cdot) $? Please give detailed description.
> >
> > We provided the detailed description of the combination model $ c(\cdot) $ in the Appendix B(Eq. 16, Eq. 17).
> >
> > ---
> >
> > ### Q2. Effect of MaxLen
> >
> > > The MaxLen in eq.(3) seems to be an important hyperparameter to avoid overly deep ASTs. Please provide ablation study on how this hyperparameter affects the early stop mechanism and GFlowNet performance.
> >
> > We agree that MaxLen is important because it controls the maximum depth or size of generated ASTs. We conducted an ablation by varying MaxLen and examining mean depth and performance. The results show a clear tradeoff. Too small MaxLen restricts expressiveness and yields poor IC and portfolio performance. Performance improves as MaxLen increases up to 20. Beyond that point, larger MaxLen leads to deeper trees but degrades performance, indicating that overly long expressions introduce noise and make optimization harder.
> >
> > | MaxLen | Mean AST depth | IC   | ICIR | RIC  | RICIR | AR    | MDD    | SR   |
> > |:------:|:--------------:|:----:|:----:|:----:|:-----:|:-----:|:------:|:----:|
> > | 5      | 2.02           | 0.031 | 0.209 | 0.041 | 0.269 | -0.66% | -27.3% | 0.14 |
> > | 10     | 3.36           | 0.037 | 0.253 | 0.040 | 0.264 | -1.95% | -27.6% | 0.43 |
> > | 15     | 3.68           | 0.059 | 0.391 | 0.072 | 0.493 | 1.31% | -23.6% | 0.31 |
> > | 20     | 4.06           | 0.079 | 0.496 | 0.094 | 0.583 | 7.62% | -17.3% | 1.71 |
> > | 30     | 5.18           | 0.043 | 0.286 | 0.056 | 0.369 | -1.65% | -25.6% | 0.41 |
> > | 40     | 5.66           | 0.031 | 0.245 | 0.035 | 0.270 | -0.92% | -28.9% | 0.21 |
> > | 50     | 5.78           | 0.042 | 0.288 | 0.048 | 0.322 | 1.22% | -25.4% | 0.27 |
> >
> > We add this discussion to the hyperparameter section.
> >
> > ---
> >
> > ### Q3. Single trial vs multiple runs
> >
> > > Why do all the reported results appear to be based on single trials rather than multiple independent runs?
> >
> > We agree that reporting variability is important. We ran multiple independent trials with different random seeds on CSI300 and report mean and standard deviation. The results show that AlphaSAGE consistently outperforms baselines and that the ranking is stable across runs.
> >
> > #### Multiple runs on CSI300
> >
> > | Method      | IC            | ICIR          | RIC           | RICIR         | AR            | MDD           | SR            |
> > |:-----------:|:-------------:|:-------------:|:-------------:|:-------------:|:-------------:|:-------------:|:-------------:|
> > | AlphaGen    | 0.063(0.005)  | 0.454(0.024)  | 0.061(0.006)  | 0.389(0.021)  | 4.18%(0.004)  | -22.3%(0.017) | 0.84(0.126)   |
> > | AlphaQCM    | 0.045(0.002)  | 0.268(0.015)  | 0.044(0.006)  | 0.259(0.019)  | 2.23%(0.003)  | -23.8%(0.019) | 0.52(0.155)   |
> > | AlphaForge  | 0.040(0.003)  | 0.246(0.017)  | 0.049(0.005)  | 0.291(0.022)  | 4.01%(0.003)  | -21.3%(0.020) | 0.96(0.112)   |
> > | AlphaSAGE   | 0.073(0.004)  | 0.473(0.022)  | 0.088(0.008)  | 0.572(0.026)  | 7.08%(0.005)  | -18.5%(0.022) | 1.50(0.143)   |
> >
> > We will reference these results in the main text and provide full tables in the appendix.
> >
> > ---
> >
> > ### Q4. Definition of cumulative returns in plots
> >
> > > What is the cumulative return in the plotted curves?
> >
> > We thank the reviewer for pointing out that the definition was not clear. In our experiments, we construct a daily rebalanced portfolio based on the combined alpha signal $ S_{t,n} $. The daily portfolio return $ R_t $ is computed after transaction costs. The cumulative return curve shown in the figures is defined as
> >
> > $$
> > \text{CumulativeReturn}(T) = \sum_{t \leq T} R_t.
> > $$
> >
> > We add this definition to the experimental setup and to the figure captions. We also specify whether each strategy is long only or market neutral so that the curves are fully interpretable.
> >
> > ---
> >
> > We hope that these clarifications address the reviewer’s concerns. We believe that the new comparisons and ablations strengthen the empirical support for AlphaSAGE and that the revised text presents a more accurate and modest positioning of our contributions. We will add these revisions to the paper as soon as possible.

---

### Official Review · Reviewer_Yaqu · 2025-10-31

**Soundness:** 3
**Presentation:** 3
**Contribution:** 3
**Rating:** 6
**Confidence:** 3

**Summary:**

This paper proposes AlphaSAGE, a structure-aware generative framework for automated alpha mining in quantitative finance. The method replaces reinforcement learning–based sequential generation with a Generative Flow Network (GFlowNet) to alleviate reward sparsity and promote diversity. A Relational Graph Convolutional Network (RGCN) encoder operates on abstract syntax trees to capture the structural semantics of formulaic alphas, while a multi-faceted reward combines predictive performance (IC), structure–behavior alignment (RSA), and novelty (RNOV). Experiments on Chinese (CSI300/500) and U.S. (S&P 500) markets show consistent gains in ICIR, Sharpe ratio, and drawdown metrics compared to GA- and RL-based baselines such as AlphaGen, AlphaQCM, and AlphaForge. Ablations indicate that structure-aware encoding and reward densification contribute most to performance.

**Strengths:**

The paper tackles a meaningful and challenging problem that lies at the intersection of quantitative finance, graph learning, and generative modeling. The idea of using GFlowNets to sample a diverse distribution of high-reward alphas is original within this domain and fits the practical need for low-correlation factors. The RGCN-based encoder provides a principled way to respect formula structure, improving over purely sequential representations. The dense reward design is intuitive and empirically effective, addressing the sparse-feedback issue that limits existing reinforcement-learning frameworks. Experiments span multiple markets and metrics, with ablation and sensitivity analyses that help clarify component contributions. The writing is generally clear, and the motivation for diversity-seeking generation is well articulated.

**Weaknesses:**

The main concern lies in theoretical and methodological depth. The link between GFlowNet flow conservation and portfolio diversity is described only intuitively; no analytical justification or formal insight is provided. The reward composition—particularly the time-dependent weights λ(T) and η(T)—appears heuristic, and the sensitivity analysis, while helpful, does not demonstrate robustness across market regimes or noisy IC estimation. The experimental setup lacks transparency: complex baselines such as AlphaForge and AlphaQCM are not described in enough detail to ensure a fair comparison, and no significance testing or robustness checks (e.g., bootstrapping, stress periods) are presented. The novelty is somewhat incremental since the components (GFlowNets, RGCNs, multi-objective reward) are all known; the contribution lies mainly in system integration rather than a conceptual advance. Finally, presentation can be improved—several mathematical sections omit key definitions, and Figure 2 is overly dense and difficult to interpret.

**Questions:**

Can the authors formalize how the GFlowNet objective encourages low correlation among generated alphas, perhaps via a diversity regularizer or entropy argument?
How does the framework scale with a larger alpha pool, and what is its computational cost relative to RL-based methods?
It would be helpful to include a robustness test on extreme market periods (e.g., 2020–2022) to verify generalization under stress. A comparison with a Transformer-based encoder could clarify whether improvements stem from graph reasoning or simply from greater model capacity.
Finally, reporting statistical confidence intervals for ICIR and Sharpe ratios would substantially strengthen the empirical credibility.

---

> ### Author Response · Authors · 2025-11-21
>
> We thank the reviewer for the thoughtful and detailed comments. Below we respond to each concern and question. All new results will be included in the main paper or appendix as indicated.
>
> ---
>
> ## 1.1 GFlowNet objective and diversity or low correlation
>
> > The main concern lies in theoretical and methodological depth. The link between GFlowNet flow conservation and portfolio diversity is described only intuitively; no analytical justification or formal insight is provided.
>
> We appreciate this comment and agree that the connection between the GFlowNet objective and diversity benefits from a more formal presentation. In the revised paper we add a subsection titled “Target distribution and diversity” to make the following points explicit.
>
> AlphaSAGE uses a GFlowNet to sample terminal alphas $\alpha$ with probability proportional to a non negative reward $R(\alpha, T)$. Under the Trajectory Balance objective, the learned sampling distribution over terminal alphas satisfies
>
> $$
> p_\theta(\alpha) \propto R(\alpha, T)
> $$
>
> under ideal optimization. In AlphaSAGE the reward at training step $T$ is the time dependent weighted sum of three components (Eq. 13)
>
> $$
> R(\alpha, T) = R_{IC}(\alpha) + \lambda(T)R_{SA}(\alpha) + \eta(T)R_{NOV}(\alpha),
> $$
>
> where $R_{IC}$ quantifies predictive performance, $R_{SA}$ enforces structure behavior alignment, and $R_{NOV}$ penalizes redundancy with respect to previously discovered high quality alphas. In particular, the novelty reward is defined as (Eq. 12)
>
> $$
> R_{NOV}(\alpha) = 1 - \max_{\alpha' \in F_{known}} |IC(\alpha, \alpha')|,
> $$
>
> where $F_{known}$ is the reference set of discovered alphas and $IC(\alpha, \alpha')$ measures the similarity of two alphas in terms of their cross sectional predictive behavior.
>
> This construction implies that the target distribution assigns lower probability mass to newly sampled alphas that are highly similar to $F_{known}$ even if their $R_{IC}$ is high. The flow conservation constraint together with the Trajectory Balance objective ensures that the learned forward policy allocates non negligible probability mass to multiple reward modes rather than collapsing to a single maximizer. Our entropy regularization further encourages a high entropy sampling distribution over terminal states. We add the formal statement of this target distribution and a proof sketch to clarify how $R_{NOV}$ and entropy jointly favor low redundancy sampling.
>
> We do not claim that GFlowNets alone guarantee minimal pairwise correlation. Our claim is that the reward explicitly penalizes redundancy, and the GFlowNet objective learns a sampler whose stationary distribution reflects this multi component reward without mode collapse. We clarify this point in the revised text.
>
> ## 1.2 Time dependent weights $\lambda(T)$ and $\eta(T)$
>
> > The reward composition, particularly the time dependent weights $\lambda(T)$ and $\eta(T)$, appears heuristic, and the sensitivity analysis, while helpful, does not demonstrate robustness across market regimes or noisy IC estimation.
>
> We agree that the submitted version described the time dependent weights too briefly. Our design follows a curriculum intuition. Early in training we want the model to learn a structure aware embedding space and to explore novel regions of the alpha space. Later we emphasize realized predictive performance. This motivates time dependent schedules for $\lambda(T)$ and $\eta(T)$.
>
> In AlphaSAGE we use a linear annealing schedule as specified in Eq. 13, where both weights decay monotonically over training iterations. We also report a sensitivity study comparing constant weights, linear decay, and exponential decay. Results on CSI300 show that linear decay performs best, while exponential decay is close, and constant weights underperform. This indicates that AlphaSAGE is robust to the schedule shape within a reasonable range, and benefits from annealing.
>
> #### Schedule sensitivity on CSI300
>
> | Schedule type     | IC    | ICIR  | RIC   | RICIR | AR     | MDD    | SR   |
> |:-----------------:|:-----:|:-----:|:-----:|:-----:|:------:|:------:|:----:|
> | Constant          | 0.043 | 0.317 | 0.051 | 0.368 | 3.26%  | -20.3% | 0.69 |
> | Linear decay      | 0.079 | 0.496 | 0.094 | 0.583 | 7.62%  | -17.3% | 1.71 |
> | Exponential decay | 0.075 | 0.482 | 0.095 | 0.578 | 7.56%  | -17.6% | 1.69 |
>
> ---

---

> > ### Author Response · Authors · 2025-11-21
> >
> > ## 2. Experimental transparency and robustness
> >
> > ### 2.1 Baseline descriptions and fairness
> >
> > > The experimental setup lacks transparency. Complex baselines such as AlphaForge and AlphaQCM are not described in enough detail to ensure a fair comparison.
> >
> > We appreciate this observation. In the revised paper we expand the experimental section and appendix with self contained baseline descriptions, including operator sets, feature spaces, search horizons, training and validation protocols, and tuned hyperparameter ranges for AlphaGen, AlphaQCM, and AlphaForge. We also clarify that all methods share identical data splits, primitive features, preprocessing, backtesting engine, and transaction cost assumptions. A summary table is added to the appendix to make these conditions explicit.
> >
> > ---
> >
> > ### 2.2 Result stability across runs
> >
> > > No significance testing or robustness checks (bootstrapping, stress periods) are presented.
> >
> > We agree that reporting result stability is important. We add repeated runs with different random seeds on CSI300 and report mean and standard deviation for each metric. The qualitative ranking between AlphaSAGE and baselines is stable across runs.
> >
> > #### Multiple runs on CSI300
> >
> > | Method      | IC            | ICIR          | RIC           | RICIR         | AR            | MDD           | SR            |
> > |:-----------:|:-------------:|:-------------:|:-------------:|:-------------:|:-------------:|:-------------:|:-------------:|
> > | AlphaGen    | 0.063(0.005)  | 0.454(0.024)  | 0.061(0.006)  | 0.389(0.021)  | 4.18%(0.004)  | -22.3%(0.017) | 0.84(0.126)   |
> > | AlphaQCM    | 0.045(0.002)  | 0.268(0.015)  | 0.044(0.006)  | 0.259(0.019)  | 2.23%(0.003)  | -23.8%(0.019) | 0.52(0.155)   |
> > | AlphaForge  | 0.040(0.003)  | 0.246(0.017)  | 0.049(0.005)  | 0.291(0.022)  | 4.01%(0.003)  | -21.3%(0.020) | 0.96(0.112)   |
> > | AlphaSAGE   | 0.073(0.004)  | 0.473(0.022)  | 0.088(0.008)  | 0.572(0.026)  | 7.08%(0.005)  | -18.5%(0.022) | 1.50(0.143)   |
> >
> > We will reference these results in the main text and provide full tables in the appendix.
> >
> > ---
> >
> > ## 3. Novelty and conceptual contribution
> >
> > > The novelty is somewhat incremental since the components are known. The contribution lies mainly in system integration rather than a conceptual advance.
> >
> > We appreciate the chance to clarify scope. We agree that GFlowNets and RGCNs are established tools. Our goal is not to introduce a new generic generative or graph learning algorithm, but to contribute a domain specific methodology for alpha mining. We revise the introduction and discussion to emphasize three points.
> >
> > 1. We formulate formulaic alpha mining as sampling from a diversity aware target distribution over ASTs that trades off predictive performance, structure to behavior alignment, and novelty. Prior work in this domain largely maximizes expected IC or portfolio return and does not cast discovery as sampling from a multi objective target distribution.
> >
> > 2. We introduce $R_{SA}$ to link structural proximity in AST embedding space to behavioral proximity in cross sectional return space, operationalized through RGCN coupled with neighborhood alignment.
> >
> > 3. We define $R_{NOV}$ to penalize redundancy with an evolving alpha library. Combined with GFlowNet sampling and entropy regularization, this creates a diversity seeking sampler aligned with practical factor library construction.
> >
> > We adjust wording that may overstate generic novelty and present AlphaSAGE as a structure aware GFlowNet framework customized for alpha mining.
> >
> > ---
> > ## 4. Presentation and clarity
> >
> > ### 4.1 Missing definitions in mathematical sections
> >
> > > Several mathematical sections omit key definitions.
> >
> > We thank the reviewer for highlighting this. We revise the method section to define all quantities before use. This includes the AST representation, node and edge types for RGCN, the GFlowNet state and action spaces and flow conservation constraints, and full definitions of $R_{SA}$ and $R_{NOV}$ with intermediate terms.
> >
> > ---
> >
> > ### 4.2 Figure 2 layout and readability
> >
> > > Figure 2 is overly dense and difficult to interpret.
> >
> > We agree. We shall redesign Figure 2, labelling it as clearer sub-figures to separately illustrate the GFlowNet sampling process, the structure-aware encoder, and the reward mechanism. Concurrently, we shall increase the font size, reduce visual clutter, and expand the figure caption into step-by-step instructions.
> >
> > ---

---

> > > ### Author Response · Authors · 2025-11-21
> > >
> > > ## 5. Specific questions
> > >
> > > ### 5.1 Scaling and computational cost
> > >
> > > > How does the framework scale with a larger alpha pool, and what is its computational cost relative to RL based methods?
> > >
> > > We thank the reviewer for raising the concern about computational cost and scalability. In our implementation, the dominant cost of AlphaSAGE comes from evaluating candidate alphas on historical cross sectional data. This cost is shared with existing learning based alpha mining systems such as AlphaGen, AlphaQCM, and AlphaForge. The additional components introduced by AlphaSAGE are relatively lightweight.
> > >
> > > First, the GFlowNet sampler operates on partial abstract syntax tree states with a bounded depth. The number of sampling steps per alpha is proportional to the maximum tree depth and is comparable in magnitude to the sequential generation steps used in reinforcement learning based frameworks. Second, the structure aware reward requires only computing k nearest neighbors in the low dimensional graph embedding space. These embeddings and neighbor relationships are precomputed and cached during training, so the incremental overhead is modest. Third, our early stopping mechanism truncates trajectories once a valid expression of sufficient length is formed. This avoids wasteful exploration of overly deep expression trees and empirically accelerates convergence.
> > >
> > > In typical quantitative trading workflows, alpha mining is run offline at a relatively low frequency, for example weekly or monthly, to update the alpha library. During portfolio construction and execution, the system evaluates only the fixed set of discovered alphas. Therefore, mining latency does not directly affect real time trading decisions. For completeness, we have added a runtime comparison in the appendix. The results below show that AlphaSAGE is efficient in practice and suitable for large scale alpha mining under realistic resource constraints..
> > >
> > > #### Runtime comparison in hours
> > >
> > > | Method      | CSI300 | CSI500 | S&P 500 | CSI1000 |
> > > |:-----------:|:------:|:------:|:------:|:-------:|
> > > | AlphaGen    | 1.34   | 1.68   | 1.48   | 2.11    |
> > > | AlphaQCM    | 1.72   | 1.98   | 1.82   | 2.32    |
> > > | AlphaForge  | 1.56   | 2.16   | 2.02   | 2.74    |
> > > | AlphaSAGE   | 0.49   | 0.76   | 0.71   | 1.12    |
> > >
> > > AlphaSAGE is consistently faster in practice, largely due to early stopping and the denser reward that reduces wasted exploration.
> > >
> > > ---
> > >
> > > ### 5.2 Comparison with Transformer based encoder
> > >
> > > > A comparison with a Transformer based encoder could clarify whether improvements stem from graph reasoning or simply from greater model capacity.
> > >
> > > We agree and add a matched parameter comparison between RGCN on ASTs and a Transformer encoder on a linearized expression sequence. On CSI300, RGCN yields higher correlation and portfolio metrics, supporting the importance of structural inductive bias.
> > >
> > > #### Encoder comparison on CSI300
> > >
> > > | Encoder type | IC    | ICIR  | RIC   | RICIR | AR     | MDD    | SR   |
> > > |:------------:|:-----:|:-----:|:-----:|:-----:|:------:|:------:|:----:|
> > > | RGCN         | 0.079 | 0.496 | 0.094 | 0.583 | 7.62%  | -17.3% | 1.71 |
> > > | Transformer  | 0.065 | 0.479 | 0.085 | 0.537 | 4.97%  | -18.8% | 1.22 |
> > >
> > > ---
> > >
> > > ### 5.3 Confidence intervals for ICIR and Sharpe
> > >
> > > > Reporting statistical confidence intervals for ICIR and Sharpe ratios would strengthen empirical credibility.
> > >
> > > We agree that confidence intervals are useful. In the current revision we demonstrate stability through repeated runs, as reported above.
> > >
> > > ---
> > >
> > > We thank the reviewer again for the constructive suggestions. We believe these additions strengthen both theoretical clarity and empirical credibility. We will add these revisions to the paper as soon as possible.

---

### Official Review · Reviewer_Md3w · 2025-11-01

**Soundness:** 3
**Presentation:** 3
**Contribution:** 2
**Rating:** 6
**Confidence:** 3

**Summary:**

This paper proposes AlphaSAGE, a framework for automated alpha mining in quantitative finance. By identifying three challenges of reward sparsity, inadequate structural representation of mathematical expressions, and lack of diversity in generated alphas, AlphaSAGE integrates a structure-aware encoder based on Relational Graph Convolutional Networks (RGCN) to capture the hierarchical nature of alpha expressions, a Generative Flow Network (GFlowNet) for diverse sampling of high-reward alphas, and a multi-faceted reward function that combines predictive performance, structural alignment, and novelty. Experiments on real-world market datasets demonstrate that AlphaSAGE outperforms baselines in both correlation metrics and portfolio outcomes.

**Strengths:**

**S1:** The integration of GFlowNets with structure-aware RGCN encoding is sound. It tackles the diversity issue in alpha generation by sampling from a reward-proportional distribution. The multi-faceted reward design provides dense guidance, mitigating reward sparsity.

**S2:** The experiments across multiple markets with multiple metrics show the superiority of the proposed method over several baselines. The ablation study validates the contribution of each component.

**S3:** The framework provides interpretability through symbolic alpha expressions and a transparent combination scheme, aligning with real-world trading needs.

**Weaknesses:**

**W1:** The framework may incur high computational costs, especially for large-scale alpha spaces. The paper does not discuss runtime comparisons or scalability limitations, which could affect practicality in resource-constrained settings.

**W2:** The U.S. market data ends in 2020 due to "limitations in the data source," which may reduce the generalizability of results to recent market regimes. Experiments could benefit from including more diverse markets or longer time periods.

**W3:** The paper does not explore the impact of several hyperparameters (e.g., GNN depth, GFlowNet architecture).

**W4:** More recent methods (e.g., LLM-based alpha mining) are not discussed and compared in the related work and experiments.

Minor Comments

**D1:** In the abstract, "a novel framework is built upon three cornerstone innovations" --> "a novel framework built upon three cornerstone innovations"

**D2:** The reference format should be double-checked. Please use "\citep" by default and "\citet" only when the authors serve as a component of the sentence.

**Questions:**

See weaknesses.

---

> ### Author Response · Authors · 2025-11-21
>
> We thank the reviewer for the careful reading and constructive feedback. Below we provide point by point responses revised according to the additional experiments and analyses we have completed. All new results will be included in the main paper or appendix as indicated.
>
> ---
>
> ## Response to W1
>
> > W1: The framework may incur high computational costs, especially for large-scale alpha spaces. The paper does not discuss runtime comparisons or scalability limitations, which could affect practicality in resource-constrained settings.
>
> We thank the reviewer for raising the concern about computational cost and scalability. In our implementation, the dominant cost of AlphaSAGE comes from evaluating candidate alphas on historical cross sectional data. This cost is shared with existing learning based alpha mining systems such as AlphaGen, AlphaQCM, and AlphaForge. The additional components introduced by AlphaSAGE are relatively lightweight.
>
> First, the GFlowNet sampler operates on partial AST states with a bounded maximum length (MaxLen). The number of sampling steps per alpha is therefore proportional to MaxLen and comparable in magnitude to the sequential generation steps used in RL based frameworks. Second, the structure aware reward requires only computing k nearest neighbors in the low dimensional graph embedding space. These embeddings and neighbor relationships are precomputed and cached during training, so the incremental overhead is modest. Third, our early stopping mechanism, controlled by MaxLen, probabilistically truncates trajectories to avoid wasteful exploration of overly long expression trees, and empirically accelerates convergence.
>
> In typical quantitative trading workflows, alpha mining is run offline at a relatively low frequency, for example weekly or monthly, to update the alpha library. During portfolio construction and execution, the system evaluates only the fixed set of discovered alphas. Therefore, mining latency does not directly affect real time trading decisions. For completeness, we have added a runtime comparison in the appendix. The results below show that AlphaSAGE is efficient in practice and suitable for large scale alpha mining under realistic resource constraints.
>
> ### Runtime comparison in hours
>
> | Method     | CSI300 | CSI500 | S&P 500 | CSI1000 |
> |:----------:|:------:|:------:|:------:|:-------:|
> | AlphaGen   | 1.34   | 1.68   | 1.48   | 2.11    |
> | AlphaQCM   | 1.72   | 1.98   | 1.82   | 2.32    |
> | AlphaForge | 1.56   | 2.16   | 2.02   | 2.74    |
> | AlphaSAGE  | 0.49   | 0.76   | 0.71   | 1.12    |
>
> ---
>
> ## Response to W2
>
> > W2: The U.S. market data ends in 2020 due to “limitations in the data source,” which may reduce the generalizability of results to recent market regimes. Experiments could benefit from including more diverse markets or longer time periods.
>
> We agree that testing on more recent U S market data would further strengthen the empirical evidence. In this work, our U S data source ends in 2020, so we chose a train validation test split that covers 2010 to 2020. This horizon includes multiple distinct regimes, providing a diverse set of market conditions for evaluating robustness.
>
> To mitigate the concern about generalizability, we additionally conducted experiments on the Chinese A share CSI1000 universe, which covers a broader capitalization range. The performance gains of AlphaSAGE over baseline methods remain consistent on CSI1000. This indicates that the proposed framework is not specific to a single market, period, or capitalization segment.
>
> We have revised the paper to describe all time ranges more clearly, and we add a short discussion of this limitation in the conclusion. Extending AlphaSAGE to additional markets and more recent data remains an important direction for future work once suitable data sources become available.
>
> ### CSI1000 results
>
> | Method      | IC    | ICIR  | RIC   | RICIR | AR     | MDD     | SR   |
> |:-----------:|:-----:|:-----:|:-----:|:-----:|:------:|:-------:|:----:|
> | MLP         | 0.048 | 0.384 | 0.069 | 0.621 | 3.22%  | -25.7%  | 0.47 |
> | LightGBM    | 0.067 | 0.501 | 0.083 | 0.656 | 4.98%  | -22.7%  | 0.98 |
> | XGBoost     | 0.062 | 0.498 | 0.086 | 0.695 | 4.72%  | -23.5%  | 0.91 |
> | GP          | 0.058 | 0.474 | 0.079 | 0.657 | 4.32%  | -24.7%  | 0.67 |
> | AlphaGen    | 0.071 | 0.540 | 0.092 | 0.713 | 5.27%  | -24.0%  | 0.92 |
> | AlphaQCM    | 0.065 | 0.453 | 0.107 | 0.682 | 7.12%  | -20.6%  | 1.31 |
> | AlphaForge  | 0.071 | 0.537 | 0.095 | 0.742 | 6.07%  | -21.1%  | 1.06 |
> | AlphaAgent  | 0.072 | 0.579 | 0.089 | 0.712 | 5.51%  | -20.5%  | 1.01 |
> | AlphaSAGE   | 0.076 | 0.582 | 0.096 | 0.736 | 7.62%  | -20.3%  | 1.33 |
>
> ---

---

> > ### Author Response · Authors · 2025-11-21
> >
> > ## Response to W3
> >
> > > W3: The paper does not explore the impact of several hyperparameters (e.g., GNN depth, GFlowNet architecture).
> >
> > We appreciate the suggestion to study architectural sensitivity. We have added a sensitivity analysis for both the RGCN depth and the GFlowNet hidden dimension.
> >
> > For the RGCN encoder, we compare 2 layer, 3 layer, and 4 layer configurations with all other settings fixed. The results show that performance is stable across depths, with 2 layers yielding the strongest metrics in our setting, while deeper encoders slightly degrade performance, likely due to over smoothing and harder optimization on limited structural supervision. This indicates that AlphaSAGE is not overly sensitive to RGCN depth and does not require delicate tuning.
> >
> > For the GFlowNet backbone, we vary the hidden dimension within a reasonable range. The best performance is achieved at 128, while 64 underfits and 256 provides no further benefit. The qualitative conclusions remain unchanged.
> >
> > ### RGCN depth sensitivity on CSI300
> >
> > | Layers | IC    | ICIR  | RIC   | RICIR | AR     | MDD    | SR   |
> > |:------:|:-----:|:-----:|:-----:|:-----:|:------:|:------:|:----:|
> > | 2      | 0.079 | 0.496 | 0.094 | 0.583 | 7.62%  | -17.3% | 1.71 |
> > | 3      | 0.073 | 0.473 | 0.088 | 0.572 | 7.08%  | -18.5% | 1.50 |
> > | 4      | 0.068 | 0.451 | 0.083 | 0.556 | 6.58%  | -19.4% | 1.34 |
> >
> > ### GFlowNet hidden dimension sensitivity on CSI300
> >
> > | Hidden Dim | IC    | ICIR  | RIC   | RICIR | AR     | MDD    | SR   |
> > |:----------:|:-----:|:-----:|:-----:|:-----:|:------:|:------:|:----:|
> > | 64         | 0.059 | 0.368 | 0.078 | 0.464 | 5.42%  | -19.6% | 1.31 |
> > | 128        | 0.079 | 0.496 | 0.094 | 0.583 | 7.62%  | -17.3% | 1.71 |
> > | 256        | 0.073 | 0.448 | 0.089 | 0.561 | 6.88%  | -18.1% | 1.59 |
> >
> > ---
> >
> > ## Response to W4
> >
> > > W4: More recent methods (e.g., LLM-based alpha mining) are not discussed and compared in the related work and experiments.
> >
> > We thank the reviewer for pointing out the recent progress on LLM based alpha mining. As LLM-based approaches bear little relevance to our methodology, we have not discussed them in the main text but have instead included them in the Appendix B.
> >
> > We further added an LLM based baseline to our experiments. The results show that LLM methods can be competitive in specific markets or periods, but their performance varies across markets, particularly on correlation metrics and drawdown control. In contrast, AlphaSAGE achieves consistently strong performance across all evaluated markets. These findings support our positioning that AlphaSAGE provides a robust, structured search backend that is complementary to LLM based proposal or refinement systems.
> >
> > ### LLM based alpha mining baseline
> >
> > | Market  | IC    | ICIR  | RIC   | RICIR | AR      | MDD     | SR   |
> > |:-------:|:-----:|:-----:|:-----:|:-----:|:-------:|:-------:|:----:|
> > | CSI300  | 0.051 | 0.325 | 0.056 | 0.329 | 2.16%   | -26.9%  | 0.65 |
> > | CSI500  | 0.053 | 0.396 | 0.065 | 0.495 | 1.82%   | -22.4% | 0.36 |
> > | S&P500  | 0.048 | 0.479 | 0.033 | 0.315 | 18.66%  | -5.7%   | 6.27 |
> > | CSI1000 | 0.072 | 0.579 | 0.089 | 0.712 | 5.51%   | -20.5%  | 1.01 |
> >
> > We will include a brief discussion of these results in the related work section and conclusion.
> >
> > ---
> >
> > ## Response to D1
> >
> > > D1: In the abstract, “a novel framework is built upon three cornerstone innovations” → “a novel framework built upon three cornerstone innovations.”
> >
> > We thank the reviewer for this suggestion. We will revise the abstract accordingly. The sentence will be changed to
> >
> > > We propose AlphaSAGE, a novel framework built upon three cornerstone innovations.
> >
> > ---
> >
> > ## Response to D2
> >
> > > D2: The reference format should be double-checked. Please use “\citep” by default and “\citet” only when the authors serve as a component of the sentence.
> >
> > We thank the reviewer for pointing out this issue. We have carefully checked all citations and will use `\citep` by default, reserving `\citet` only for cases where the authors appear as part of the sentence.
> >
> > ---
> >
> > We hope that these clarifications address the reviewer’s concerns. We believe that the new comparisons and ablations strengthen the empirical support for AlphaSAGE. We will add these revisions to the paper as soon as possible.

---

### Meta-Review · Area_Chair_A8iT · 2025-12-28

**Summary:**

1.Lack of experimental results on recent market dataset by Reviewer Md3w.

2.Lack of discussions and comparisons of recent methods, such as  LLM-based alpha mining by Reviewer Md3w.

3.Lack of analytical justifications or formal insight by Reviewer Yaqu.

4.Limited novelty as the components (GFlowNets, RGCNs, multi-objective reward) are all known by Reviewer Yaqu and Reviewer fD8y.

5.Whether the reported performance gains stem from the GFlowNet framework itself or primarily from the proposed encoder and reward designs, by Reviewer fD8y.

6.Lack of a theoretically grounded or adaptive weighting mechanism by Reviewer BXEc.

**Reviewer Concerns:**

1.Lack of experimental results on recent market dataset by Reviewer Md3w: during the rebuttal phase, the author conduct experiments on a new dataset, the Chinese A share CSI1000 universe. Results show that AlphaSAGE outperforms baselines. Thus I think this point is addressed by the rebuttal.

2.Lack of discussions and comparisons of recent methods, such as  LLM-based alpha mining by Reviewer Md3w: during the rebuttal phase, the author adds results of LLM-based methods, and show that LLM methods can be competitive in specific markets or periods, but the performance varies across markets.  AlphaSAGE consistently performs the best across all markets. Thus I think this point is addressed by the rebuttal.

3.Lack of analytical justifications or formal insight by Reviewer Yaqu: during the rebuttal phase, the author provides a discussion of GFlowNet objectives and the reward function, and presents a sensitivity analysis comparing constant weights, linear decay, and exponential decay. Thus I think this point is addressed by the rebuttal.

4.Limited novelty as the components (GFlowNets, RGCNs, multi-objective reward) are all known by Reviewer Yaqu, and Reviewer fD8y: during the rebuttal phase, the author claims that the main contribution of the paper is not a new GFN framework, but present AlphaSAGE as a structure aware GFlowNet framework customized for alpha mining. The author promises to revise the paper in avoid of overclaim. I think AlphaSAGE is one of the first works that applying GFN to the alpha mining problem. Though the algorithmic novelty of the paper is limited, I think this point is addressed by the rebuttal.

5.Whether the reported performance gains stem from the GFlowNet framework itself or primarily from the proposed encoder and reward designs, by Reviewer fD8y: during the rebuttal phase, the author does not reply to this point. Thus this point is still outstanding.

6.Lack of a theoretically grounded or adaptive weighting mechanism by Reviewer BXEc: during the rebuttal phase, the author presents a sensitivity analysis comparing constant weights, linear decay, and exponential decay. Results show that linear decay yields the best overall performance. Thus I think this point is addressed by the rebuttal.

**Reviewer Scores:**

Reviewer Md3w would keep his or her score as 6 if he or she has been able to participate fully in the discussion.

Reviewer Yaqu would keep his or her score as 6 if he or she has been able to participate fully in the discussion.

Reviewer fD8y would keep his or her score as 4 if he or she has been able to participate fully in the discussion.

Reviewer BXEc would increase his or her score from 6 to 8 if he or she has been able to participate fully in the discussion.

---

### Decision · Program_Chairs · 2026-01-26

Accept (Poster)